# Learning Image Priors through Patch-based Diffusion Models for Solving Inverse Problems

**Jason Hu**    **Bowen Song**    **Xiaojian Xu**    **Liyue Shen**    **Jeffrey A. Fessler**

Department of Electrical and Computer Engineering
University of Michigan
Ann Arbor, MI 48109
`{jashu, bowenbw, xjxu, liyues, fessler}@umich.edu`

## Abstract

Diffusion models can learn strong image priors from underlying data distributions and use them to solve inverse problems, but the training process is computationally expensive and requires lots of data. Such bottlenecks prevent most existing works from being feasible for high-dimensional and high-resolution data such as 3D images. This paper proposes a method to learn an efficient data prior for the entire image by training diffusion models only on patches of images. Specifically, we propose a patch-based position-aware diffusion inverse solver, called PaDIS, where we obtain the score function of the whole image through scores of patches and their positional encoding and use this as the prior for solving inverse problems. We show that this diffusion model achieves improved memory efficiency and data efficiency while still maintaining the ability to generate entire images via positional encoding. Additionally, the proposed PaDIS model is highly flexible and can be paired with different diffusion inverse solvers (DIS). We demonstrate that the proposed PaDIS approach enables solving various inverse problems in both natural and medical image domains, including CT reconstruction, deblurring, and superresolution, given only patch-based priors. Notably, PaDIS outperforms previous DIS methods trained on entire image priors in the case of limited training data, demonstrating the data efficiency of our proposed approach by learning patch-based prior. Code: `https://github.com/sundeco/PaDIS`

## 1 Introduction

Diffusion models learn the prior of an underlying data distribution and can use the prior to generate new images [1–3]. By starting with a clean training images and gradually adding higher levels of noise, eventually obtaining images that are indistinguishable from pure noise, the score function of the image distribution, denoted $s(x) = \nabla \log p(\boldsymbol{x})$, can be learned by a neural network. The reverse process (sampling or generation) then starts with pure noise and uses the learned score function to iteratively remove noise, ending with a clean image sampled from the underlying distribution $p(\boldsymbol{x})$.

Inverse problems are ubiquitous in image processing, and aim to reconstruct an image $\boldsymbol{x}$ from a measurement $\boldsymbol{y}$, where $\boldsymbol{y} = \mathcal{A}(\boldsymbol{x}) + \boldsymbol{\epsilon}$, $\mathcal{A}$ represents a forward operator, and $\boldsymbol{\epsilon}$ represents random unknown noise. By Bayes rule, $p(\boldsymbol{x}|\boldsymbol{y})$ is proportional to $p(\boldsymbol{x}) \cdot p(\boldsymbol{y}|\boldsymbol{x})$. Hence, to recover $\boldsymbol{x}$, it is important to have a good estimate of the prior $p(\boldsymbol{x})$, particularly when $\boldsymbol{y}$ contains far less information than $\boldsymbol{x}$. Diffusion models are known for their ability to learn a strong prior, so there is a growing literature on using them to solve inverse problems [4–8].

38th Conference on Neural Information Processing Systems (NeurIPS 2024).

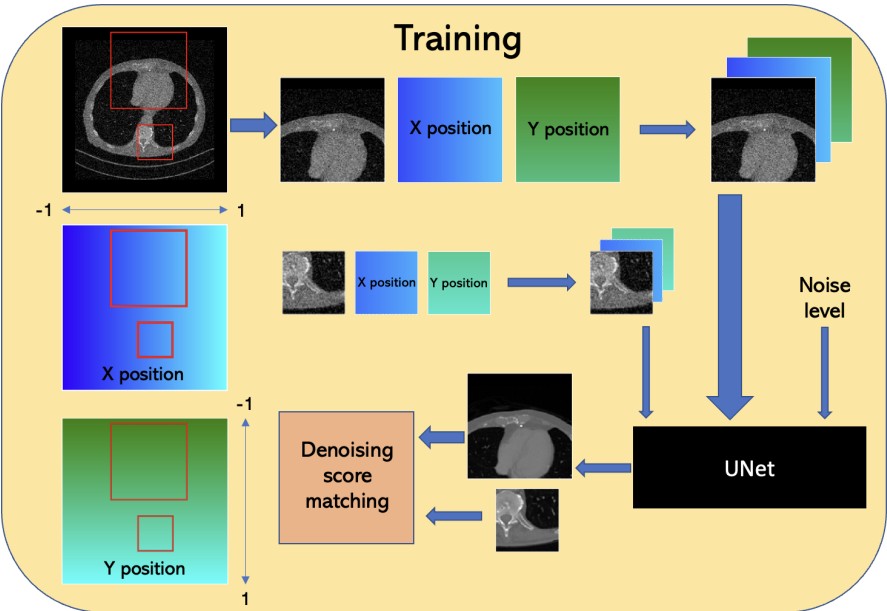

Figure 1: Training the proposed Patch Diffusion Inverse Solver (PaDIS) method. Different sized patches are used in each training iteration. The X and Y position arrays have the same size as the patch and consist of the normalized X and Y coordinates of each pixel of the patch.

However, diffusion models require large quantities of training data and vast computational power to be able to generate high resolution images; Song et al. [2] and Ho et al. [3] used several days to weeks of training on over a million training images in the ImageNet [9] and LSUN [10] datasets to generate $256 \times 256$ images. This high cost motivates the research on improved training efficiency for diffusion models, such as fine-tuning an existing checkpoint on a different dataset [11, 12] to reduce the required training time and data. However, this strategy restricts the range of usable network architectures and requires the existence of a pretrained network, which limits the range of applications. Besides the demanding training data and computational cost, diffusion models also struggle in very large-scale problems, such as very high resolution images or 3D images. To address these challenges, latent diffusion models [13, 14] have been proposed to learn the image distribution in a smaller latent space, but it is difficult to solve general inverse problems in the latent space [15]. Patch-based diffusion models have also been proposed to reduce computational cost. For example, Wang et al. [16] trained on patches of images, but for image generation, ultimately still relied on computing the score function of the whole image at once. Ding et al. [17] used patches in the feature space, requiring an additional encoding and decoding step. For 3D volumes, the most common method involves breaking up the volume into 2D slices [12], [18]. These methods add regularizers between slices to enforce consistency during sampling, and thus do not provide a self-contained method for computing the score function of the whole volume. These application-specific strategies make it difficult to adapt these methods to general purpose inverse problem solvers using diffusion models [5], [19], [7], [20].

Our proposed method tackles these challenges in a unified way by training diffusion models on patches of images, as opposed to whole images (see Fig. 1). We provide the location of the randomly extracted patch to the network to help it learn global image properties. Since each training image contains many patches, the required size of the training dataset is greatly reduced, from the millions usually needed to generate high quality images to only a couple thousand or even several hundred (see Tab. 5). The required memory and training time is also reduced because it is never necessary to backpropagate the whole image through the network. Our proposed method allows for a flexible network architecture and only requires it to accept images of any size, a property true of many UNets [3], so there is much more flexibility in the architecture design than fine-tuning methods.

At inference time (see Fig. 2), by first zero padding the image, the proposed approach partitions it into patches in many different ways (see Fig. 3), eliminating the boundary artifacts between patches that would appear if non-overlapping patches were used. We develop a method to express the distribution of the whole image in terms of the patch distribution that is learned by the proposed patch-based

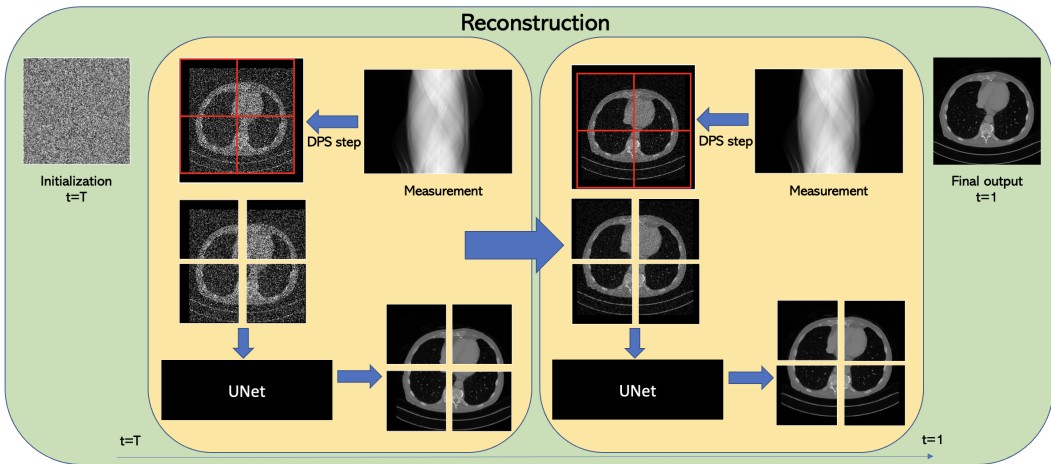

Figure 2: Overview of reconstruction process for the proposed Patch Diffusion Inverse Solver (PaDIS) method. Starting at $t = T$, at each iteration we choose a random partition of the zero padded image and use the neural network trained on patches to get the score function of the entire image. Due to the shifting patch locations, the output image has no boundary artifacts.

network. By incorporating positional information of patches, this framework allows us to compute the score function of the whole image without ever inputting the whole image into the network. Unlike previous patch-based works that may be task-specific [21–23], the prior defined by our approach may be treated in a black box manner and then paired with any other stochastic differential equation (SDE) solver to sample from the prior, or with any general purpose inverse problem solver to perform image reconstruction. We conduct experiments on multiple datasets and different inverse problems and demonstrate that the proposed method is able to synthesize the patches to produce reasonably realistic images and very accurate reconstructions for inverse problems. Furthermore, PaDIS provides a promising avenue for which generation and inverse problem solving of very large and high dimensional images may be tackled in the future.

In summary, our main contributions are as follows:

- We provide a theoretical framework whereby a score function of a high-resolution high-dimensional image is learned purely through the score function of its patches.
- The proposed method greatly reduces the amount of memory and training data needed compared to traditional diffusion models.
- The trained network has great flexibility and can be used with many existing sampling algorithms and is the first patch-based model that can solve inverse problems in an unsupervised manner.
- We perform experiments on a variety of inverse problems to show superior image quality over whole image diffusion model methods while being far less resource heavy.

## 2 Background and Related Work

**Diffusion models.** Diffusion models consist of defining a forward stochastic differential equation (SDE) that adds noise to a clean image [2]: for $t \in [0, T]$, $\boldsymbol{x}(t) \in \mathbb{R}^d$, we have

$$\mathrm{d}\boldsymbol{x} = -(\beta(t)/2)\,\boldsymbol{x}\,\mathrm{d}t + \sqrt{\beta(t)}\,\mathrm{d}\boldsymbol{w}, \qquad (1)$$

where $\beta(t)$ is the noise variance schedule of the process. The distribution of $\boldsymbol{x}(0)$ is the data distribution and the distribution of $\boldsymbol{x}(T)$ is (approximately) a standard Gaussian. Then, image generation is done through the reverse SDE [24]:

$$\mathrm{d}\boldsymbol{x} = \left(-\beta(t)/2 - \beta(t)\nabla_{\boldsymbol{x}_t} \log p_t(\boldsymbol{x}_t)\right)\,\mathrm{d}t + \sqrt{\beta(t)}d\overline{\boldsymbol{w}}. \qquad (2)$$

By training a neural network to learn the score function $\nabla_{\boldsymbol{x}_t} \log p_t(\boldsymbol{x}_t)$, one can start with noise and run the reverse SDE to obtain samples from the learned data distribution.

To reduce the computational burden, latent diffusion models [13] have been proposed, aiming to perform the diffusion process in a much smaller latent space, allowing for faster training and sampling. However, that method requires a pretrained encoder and decoder [25] for a fixed dataset, so it must be retrained for different datasets, and it still requires large amounts of training data. Patch-based diffusion models [16, 17] focus on image generation while training only on patches. Supervised patch-based diffusion methods [23, 26] are task specific and do not learn an unconditional image prior that can be applied to all inverse problems. Other patch-based methods [27–29] learn an unconditional image prior but require the whole image as an input during inference time. Finally, work has been done to perform sampling faster [14, 30, 31], which is unrelated to the training process.

**Solving inverse problems.** For most real-world inverse problems, the partial measurement $y$ is corrupted and incomplete, so the mapping from $x$ to $y$ is many-to-one, even in the absence of noise, making it impossible to exactly recover $x$. Hence, it is necessary to enforce a prior on $x$. Traditionally, methods such as total variation (TV) [32] and wavelet transform [33] have been used to enforce image sparsity. To capture more global image information, low-rank methods are also popular [34–37]. These methods frequently involve solving an optimization problem that simultaneously enforces data consistency and the image prior.

In recent years, data-driven methods have risen in popularity in signal and image processing [38–42]. In particular, for solving inverse problems, when large amounts of training data is available, a learned prior can be much stronger than the handcrafted priors used in traditional methods. For instance, plug and play and regularized by denoising methods [43–49] involve pretraining a denoiser and applying it at reconstruction time to iteratively recover the image. These methods have the advantage over supervised deep learning methods such as [50–53] that the same denoiser may be applied to solve a wide variety of inverse problems.

Diffusion models serve as an even stronger prior as they can generate entire images from pure noise. Most methods that use diffusion models to solve inverse problems involve writing the problem as a conditional generation problem [54–56] or as a posterior sampling problem [4–7, 57]. In the former case, the network requires the measurement $y$ (or an appropriate resized transformation of $y$) during training time. Thus, for these task-specific trained models, at reconstruction time, that network is useful only for solving that specific inverse problem. In contrast, for the posterior sampling framework, the network learns an unconditional image prior for $x$ that can help solve any inverse problem related to $x$ without retraining. Our proposed method may be paired with most existing posterior sampling algorithms [5–7, 58].

## 3  Methods

To be able to solve large 2D imaging problems as well as 3D and 4D inverse problems, our goal is to develop a model for $p(x)$ that does not require inputting the entire image $x$ into any neural network. We would like to simply partition $x$ into nonoverlapping patches, learn the distribution of each of the patches, and then piece them together to obtain the distribution of $x$. However, this would result in boundary artifacts between the patches. Directly using overlapping patches would result in sections of the image covered by multiple patches to be updated multiple times, which is inconsistent with the theory of diffusion models. Ideally, we would like to use nonoverlapping patches to update $x$, but with a variety of different patch tiling schemes so that boundaries between patches do not remain the same through the entire diffusion process.

To accomplish this task, we zero pad $x_0$ and learn the distribution of the resulting zero padded image. More precisely, consider an $N \times N$ image $x_0$ and let $1 \leq P < N$ be an integer denoting the patch size and let $k = \lfloor N/P \rfloor$. Then $x_0$ could be covered with a $(k+1) \times (k+1)$ nonoverlapping patch grid but that would result in $(k+1)P - N$ additional rows and columns for the patches. Hence, we zero pad $x_0$ on all four sides by $M = (k+1)P - N$ to form a new image with $N + 2M$ rows and columns. With slight abuse of notation, we also denote this larger image by $x$. Let $i, j$ be positive integers between 1 and $M$ inclusive. Fig. 3 illustrates that we may partition $x$ into $(k+1)^2 + 1$ regions as follows: $(k+1)^2$ of the regions consist of evenly chopping up the square consisting of rows $i$ through $i + N + P$ and columns $j$ through $j + N + P$ into a $k+1$ by $k+1$ grid, with each such partition being $P \times P$, and the last region consists of the remaining bordering part of $x$ that is not included in the first $(k+1)^2$ regions. This last region will always be entirely zeros, and the $(k+1)^2$ square patches fully cover the central $N \times N$ image.

Each pair of integers $i$ and $j$ correspond to one possible partition, so when we let $i$ and $j$ range through all the possible values, our proposal is to model the distribution of $\boldsymbol{x}$ as the following product distribution:

$$p(\boldsymbol{x}) = \prod_{i,j=1}^{i,j=M} p_{i,j,B}(\boldsymbol{x}_{i,j,B}) \prod_{r=1}^{(k+1)^2} p_{i,j,r}(\boldsymbol{x}_{i,j,r})/Z, \tag{3}$$

where $\boldsymbol{x}_{i,j,B}$ represents the aforementioned bordering region of $\boldsymbol{x}$ that depends on the specific values of $i$ and $j$, $p_{i,j,B}$ is the probability distribution of that region, $\boldsymbol{x}_{i,j,r}$ is the $r$th $P \times P$ patch when using the partitioning scheme corresponding to the values of $i$ and $j$, $p_{i,j,r}$ is the probability distribution of that region, and $Z$ is an appropriate scaling factor. Generative models based on products of patch probabilities have a long history in the Markov random field literature; see §A.6.

The score function of the entire image follows directly from (3):

$$\nabla \log p(\boldsymbol{x}) = \sum_{i,j=1}^{i,j=M} \left( \boldsymbol{s}_{i,j,B}(\boldsymbol{x}_{i,j,B}) + \sum_{r=1}^{(k+1)^2} \boldsymbol{s}_{i,j,r}(\boldsymbol{x}_{i,j,r}) \right). \tag{4}$$

Thus, we have expressed the score function of the whole image $\boldsymbol{x}$ as sums of scores of patches $\boldsymbol{x}_{i,j,r}$ and the border $\boldsymbol{x}_{i,j,B}$. The former can be learned through score matching as in §3.1. For the latter, by construction, if $\boldsymbol{x}$ is a zero padded image then $\boldsymbol{x}_{i,j,B}$ is everywhere zero. Thus, for all $\boldsymbol{x}$, $D(\boldsymbol{x}_{i,j,B}) = \mathbb{E}[\boldsymbol{x}_{i,j,B}, t = 0 | \boldsymbol{x}_{i,j,B}, t = T]$ is everywhere zero, where $D$ represents the denoiser function. Then the computation of its score function is trivial by Tweedie's formula [59].

Importantly, unlike previous papers like [16] and [17], our method can compute the score function of the entire image through only inputs of patches to the neural network. This makes it possible to learn a black box function for score functions of large images, where directly training a diffusion model would be infeasible due to memory and time constraints. Furthermore, §4 shows that in the case of limited data, the large number of patches present in each training image allows the patch-based model to learn a model for the underlying distribution that performs better than whole image models.

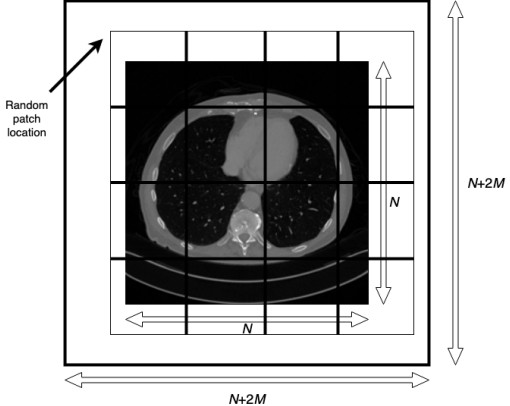

Figure 3: Schematic for zero padding and partitioning image into patches

### 3.1 Patch-wise training

Following the work in [16] and [14], we train a denoiser using the score matching approach. Our neural network $D_\theta(\boldsymbol{x}, \sigma_t)$ accepts the noisy image $\boldsymbol{x}$ and a noise level $\sigma_t$, and is trained through the following loss function:

$$\mathbb{E}_{t \sim \mathcal{U}(0,T)} \mathbb{E}_{\boldsymbol{x} \sim p(\boldsymbol{x})} \mathbb{E}_{\boldsymbol{\epsilon} \sim \mathcal{N}(0, \sigma_t^2 I)} \| D_\theta(\boldsymbol{x} + \boldsymbol{\epsilon}, \sigma_t) - \boldsymbol{x} \|_2^2. \tag{5}$$

By Tweedie's formula [59], the score function is given by $s_\theta(\boldsymbol{x}, \sigma_t) = (D_\theta(\boldsymbol{x}, \sigma_t) - \boldsymbol{x})/\sigma_t^2$. Here, we want to learn the score function of patches $\boldsymbol{x}_{i,j,r}$, so we apply (5) to patches of $\boldsymbol{x}$ instead of the entire image. Following [16], we extract patches randomly from the zero-padded image $\boldsymbol{x}$. To incorporate the positional information of the patch, we first define the $x$ positional array as the size $N + 2M$ 2D array consisting of the $x$ positions of each pixel of the image scaled between -1 and 1; the $y$ positional array is similarly defined. We then extract the corresponding patches of these two positional arrays and concatenate them along the channel dimension of the noisy image patch as inputs into the network, nothing that noise is not added to the positional arrays.

When directly applying (5), it would suffice to fix the patch size $P$ and then train using size $P$ patches exclusively. However, [16] found that by training with patches of varying sizes, training time can be reduced and the neural network learns cross-region dependencies better. Hence, we train both with patches of size $P$ and also patches of *smaller* size, where the size is chosen according to a stochastic scheduling scheme. By using the UNet architecture in [3], the same network can take images of different sizes as input. The Appendix provides details of the experiments.

## 3.2 Sampling and reconstruction

The proposed patch-based method for learning $p(\boldsymbol{x})$ may be paired with any method that would otherwise be used for sampling with a full image diffusion model, such as Langevin dynamics [1] and DDPM [3], as well as acceleration methods such as second-order solvers [14] and DDIM [30]. At training time, the network only receives patches of the image as input, along with the positional information of the patch. Nevertheless, we show that when the number of sampling iterations is sufficiently large, the proposed method is still capable of generating reasonably realistic appearing entire images. However, our main goal is solving large-scale inverse problems, not image generation.

Computing $\boldsymbol{s}(\boldsymbol{x})$ via (4) would require summing over the score functions of all $(k+1)^2$ patches a total of $M^2$ times (corresponding to the $M^2$ different ways of selecting $i$ and $j$). This method would be prohibitively slow due to the size of $M^2$; instead, for each iteration we randomly choose integers $i$ and $j$ between 1 and $M$ and estimate $\boldsymbol{s}(\boldsymbol{x})$ using just that corresponding term of the outer summation.

For solving inverse problems with diffusion models, there are general algorithms e.g., [19] and [5], as well as more model-specific algorithms, e.g., [6] and [7]. Here, we demonstrate that our method applies to a broad range of inverse problems, and opt to use generalizable methods that do not rely on any special properties (such as the singular value decomposition of the system matrix as in [6], [7]) of the forward operator. We found that DPS [5] in conjunction with Langevin dynamics generally yielded the most stable and high quality results, so we use this approach as our central algorithm. Similar to [5], we chose the stepsize to be $\zeta_i = \zeta / \|\boldsymbol{y} - \mathcal{A}(D(\boldsymbol{x}))\|_2$. To the best of our knowledge, this is the first work that learns a fully patch-based diffusion prior and applies it to solve inverse problems; we call our method **Pa**tch **D**iffusion **I**nverse **S**olver (PaDIS). Computing the score functions of the patches at each iteration is easily parallelized, allowing for fast

---

**Algorithm 1** Patch Diffusion Inverse Solver (PaDIS)

---

**Require:** $\sigma_1 < \sigma_2 < \ldots < \sigma_T$, $\epsilon > 0$, $\zeta_i > 0$, $P, M, \boldsymbol{y}$
    Initialize $\boldsymbol{x} \sim \mathcal{N}(0, \sigma_T^2 \boldsymbol{I})$
    **for** $t = T : 1$ **do**
        Sample $z \sim \mathcal{N}(0, \sigma_t^2 \boldsymbol{I})$
        Set $\alpha_t = \epsilon \cdot \sigma_t^2$
        Randomly select integers $i, j \in [1, M]$
        For all $1 \leq r \leq (k+1)^2$, extract patch $\boldsymbol{x}_{i,j,r}$
        Compute $D_{i,j,r} = D_\theta(\boldsymbol{x}_{i,j,r}, \sigma_t)$
        Set $\boldsymbol{s}_{i,j,r} = (D_{i,j,r} - \boldsymbol{x}_{i,j,r})/\sigma_t^2$
        Apply (4) to get $\boldsymbol{s} = \boldsymbol{s}(\boldsymbol{x}, \sigma_t)$
        Set $\boldsymbol{x}$ to $\boldsymbol{x} - \zeta_t \nabla_{\boldsymbol{x}_t} \|\boldsymbol{y} - \mathcal{A}(D)\|_2^2$
        Set $\boldsymbol{x}$ to $\boldsymbol{x} + \frac{\alpha_t}{2}\boldsymbol{s} + \sqrt{\alpha_t}\boldsymbol{z}$
    **end for**

---

sampling and reconstruction. Alg. 1 shows the pseudocode for the main image reconstruction algorithm; the appendix shows the pseudocode for the other implemented algorithms.

Finally, we comment on some high-level similarities between our proposed method and [18]; in both cases, the image in question is partitioned into smaller parts in multiple ways. In [18], one of the partitions consists of 2D slices in the x-y direction, and the other partition consists of 2D slices in the x-z direction, whereas for our method, each of the partitions consists of $(k+1)^2$ square patches and the outer border region. For both methods, the score functions of each of the parts are learned independently during training. Then for each sampling iteration, both methods involve choosing one of the partitions, computing the score functions for each of the parts independently, and then updating the entire image by updating the parts separately. For our approach, the zero-padding method allows for many possible partitions of the image and eliminates boundary artifacts between patches.

## 4 Experiments

**Experimental setup.** For the main CT experiments, we used the AAPM 2016 CT challenge data [60] that consists of 10 3D volumes. We cropped the data in the Z-direction to select 256 slices and then rescaled the images in the XY-plane to have size $256 \times 256$. Finally, we used the XY slices from 9 of the volumes to define 2304 training images, and used 25 of the slices from the tenth volume as test data. For the deblurring and superresolution experiments, we used a 3000 image subset of the CelebA-HQ dataset [61] (with each image being of size $256 \times 256$) for training to demonstrate that the proposed method works well in cases with limited training data. We preprocessed the data by dividing all the RGB values by 255 so that all the pixel values were between 0 and 1. The test data was a randomly selected subset of 25 of the remaining images. In all cases, we report the average

metrics across the test images: peak SNR (PSNR) in dB, and structural similarity metric (SSIM) [62]. For the colored images, these metrics were computed in the RGB domain.

For the main patch-based networks, we trained mostly with a patch size of $56 \times 56$ to allow the target image to be completely covered by 5 patches in each direction while minimizing the amount of zero padding needed. We used the network architecture in [14] for both the patch-based networks and whole image networks. All networks were trained on PyTorch using the Adam optimizer with 2 A40 GPUs. The Appendix provides full details of the hyperparameters.

**Image generation.** Our proposed method is able to learn a reasonable prior for whole images, despite never being trained on any whole images. Fig. 4 shows generation results for the CT dataset using three different methods. The top row used the network trained on whole images; the middle row used the method of [16] except that the entire image is never supplied to the network either at training or sampling time; the bottom row used the proposed method. The middle row shows that the positional encoding does ensure reasonably appropriate generated patches at each location. However, simply generating each of the patches independently and then naively assembling them together leads to obvious boundary artifacts due to lack of consistency between patches. Our proposed method solves this problem by using overlapping patches via random patch grid shifts, leading to generated images with continuity throughout.

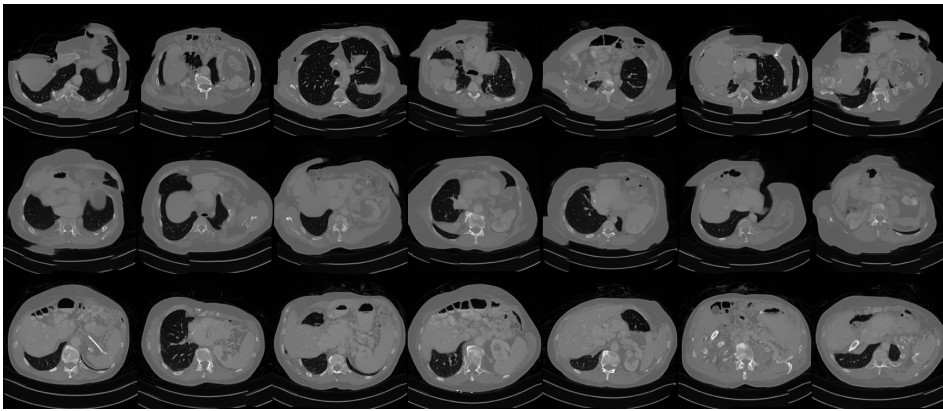

Figure 4: Unconditional generation of CT images. Top row: generation with a network trained on whole image; middle row: patch-only version of [16]; bottom row: proposed PaDIS method.

**Inverse problems.** We tested the proposed method on a variety of different inverse problems: CT reconstruction, deblurring, and superresolution. For the forward and backward projectors in CT reconstruction, we used the implementation provided by [63]. We performed two sparse view CT (SVCT) experiments: one using 8 projection views, and one using 20 projection views. Both of these were done using a parallel-beam forward projector where the detector size was 512 pixels. For the deblurring experiments, we used a uniform blur kernel of size $9 \times 9$ and added white Gaussian noise with $\sigma = 0.01$ where the clean image was scaled between 0 and 1. For the superresolution experiments, we used a scaling factor of 4 with downsampling by averaging and added white Gaussian noise with $\sigma = 0.01$. DPS has shown to benefit significantly from using a higher number of neural function evaluations (NFEs) [5], so we use 1000 NFEs for all of the diffusion model experiments. Appendix A.7 discusses this further.

For the comparison methods, we trained a diffusion model on entire images using the same denoising score matching method shown in Section 3.1. The inference process was identical to that of the patch-based method, with the exception that the score function of the image at each iteration was computed directly using the neural network, as opposed to needing to break up the zero-padded image into patches. We also compared with two traditional methods: applying a simple baseline and reconstructing via the total variation regularizer (ADMM-TV). For CT, the baseline was obtained by applying the filtered back-projection method to the measurement $\boldsymbol{y}$. For deblurring, the baseline was simply equal to the blurred image. For superresolution, the baseline was obtained by upsampling the low resolution image and using nearest neighbor interpolation. The implementation of ADMM-TV can be found in [64]. We also implemented two plug and play (PnP) methods: PnP-ADMM [42]

and regularization by denoising (RED) [46]. We trained denoising CNNs on each of the datasets following [65] and then used them to solve the inverse problems.

For further comparison of diffusion model methods, we implemented different sampling and data consistency algorithms and applied them in conjunction with our patch-based prior. In particular, we compared with Langevin dynamics [1], predictor-corrector sampling [19], and variation exploding DDNM (VE-DDNM) [7]. For all these sampling methods, we used the variation exploding framework for consistency and to avoid retraining the network. We also compared with two other ways of assembling priors of patches to form the prior of an entire image: patch averaging [23] and patch stitching [66]. App. A.5 contains pseudocode for these comparison algorithms.

Table 1 shows the main inverse problem solving results. The best results were obtained after training the patch-based models for around 12 hours, while the whole image models needed to be trained for 24-36 hours, demonstrating a significant improvement in training time. Furthermore, when averaging the metrics across the test dataset, our proposed method outperformed the whole image method in terms of PSNR and SSIM for all the inverse problems. The score functions of all the patches can be computed in parallel for each iteration, so the reconstruction times for these methods were very similar (both around 5 minutes per image). The whole-image results could be more favorable if more training data were used. See data-size study in App. A.3.

We also ran ablation studies examining the effect of various parameters of the proposed method. Namely, we studied the usage of different patch sizes during reconstruction, varying dataset sizes, importance of positional encoding for patches, and different sampling and inverse problem solving algorithms. The results of these studies are in App. A.2.

Table 1: Comparison of quantitative results on three different inverse problems. Results are averages across all images in the test dataset. Best results are in bold.

| Method | CT, 20 Views | | CT, 8 Views | | Deblurring | | Superresolution | |
|---|---|---|---|---|---|---|---|---|
| | PSNR↑ | SSIM↑ | PSNR↑ | SSIM↑ | PSNR↑ | SSIM↑ | PSNR↑ | SSIM↑ |
| Baseline | 24.93 | 0.595 | 21.39 | 0.415 | 24.54 | 0.688 | 25.86 | 0.739 |
| ADMM-TV | 26.82 | 0.724 | 23.09 | 0.555 | 28.22 | 0.792 | 25.66 | 0.745 |
| PnP-ADMM [42] | 26.86 | 0.607 | 22.39 | 0.489 | 28.82 | 0.818 | 26.61 | 0.785 |
| PnP-RED [46] | 27.99 | 0.622 | 23.08 | 0.441 | 29.91 | 0.867 | 26.36 | 0.766 |
| Whole image diffusion | 32.84 | 0.835 | 25.74 | 0.706 | 30.19 | 0.853 | 29.17 | 0.827 |
| Langevin dynamics [1] | 33.03 | 0.846 | 27.03 | 0.689 | 30.60 | 0.867 | 26.83 | 0.744 |
| Predictor-corrector [19] | 32.35 | 0.820 | 23.65 | 0.546 | 28.42 | 0.724 | 26.97 | 0.685 |
| VE-DDNM [7] | 31.98 | 0.861 | 27.71 | 0.759 | - | - | 26.01 | 0.727 |
| Patch Averaging [23] | 33.35 | 0.850 | 28.43 | 0.765 | 29.41 | 0.847 | 27.67 | 0.802 |
| Patch Stitching [66] | 32.87 | 0.837 | 26.71 | 0.710 | 29.69 | 0.849 | 27.50 | 0.780 |
| PaDIS (Ours) | **33.57** | **0.854** | **29.48** | **0.767** | **30.80** | **0.870** | **29.47** | **0.846** |

In addition to the main inverse problems shown in Table 1, we also ran experiments on three additional inverse problems to demonstrate the effectiveness of our method on a wide variety of inverse problems: 60 view CT reconstruction, fan beam reconstruction using 180 views, and deblurring with a uniform kernel of size $17 \times 17$. For the deblurring experiment, we again added Gaussian noise with level $\sigma = 0.01$ to the measurement. Table 2 shows the quantitative results of these experiments and Figure A.10 shows the visual results.

Table 2: Extra inverse problem experiments. Results are averages across all images in the test dataset. Best results are in bold.

| Method | CT, 60 Views | | CT, Fan Beam | | Heavy Deblurring | |
|---|---|---|---|---|---|---|
| | PSNR↑ | SSIM↑ | PSNR↑ | SSIM↑ | PSNR↑ | SSIM↑ |
| Baseline | 25.89 | 0.746 | 20.07 | 0.521 | 21.14 | 0.569 |
| ADMM-TV | 30.93 | 0.833 | 25.78 | 0.719 | 26.03 | 0.724 |
| Whole image diffusion | 35.83 | 0.894 | 26.89 | 0.835 | 28.35 | 0.808 |
| PaDIS (Ours) | **39.28** | **0.941** | **29.91** | **0.932** | **28.91** | **0.818** |

In the bottom of Figure 5, some artifacts are present in the reconstructions obtained by the diffusion model methods, although they are more apparent in the whole image model than with PaDIS. The measurements are very compressed in this case, so it is very difficult for any model to obtain diagnostic-quality reconstructions; the baselines perform significantly worse in terms of quantitative

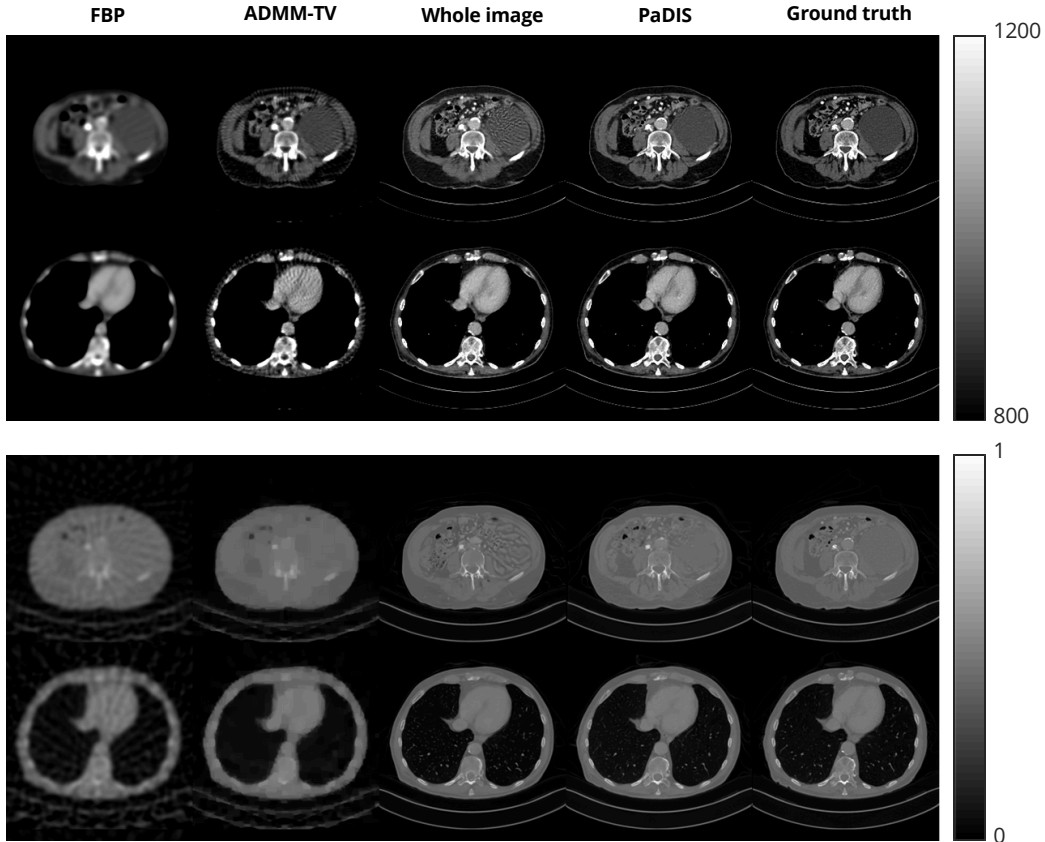

Figure 5: Results of CT reconstruction. 60 views are used for the top two rows, 20 views are used for the bottom two rows. To better show contrast between organs, we use modified Hounsfield units (HU) in the top figure, while we use the same scale the images were trained on in the bottom figure.

metrics and exhibit severe blurring. In clinical settings, patient diagnosis are typically performed with CT scans consisting of hundreds of views. The top of Figure 5 shows that when 60 views are used, our proposed method yields a much better reconstruction without artifacts. Nevertheless, we show the potential of our proposed methods to reconstruct images from very sparse views with a decent image quality, which could be potentially used for applications such as patient positioning.

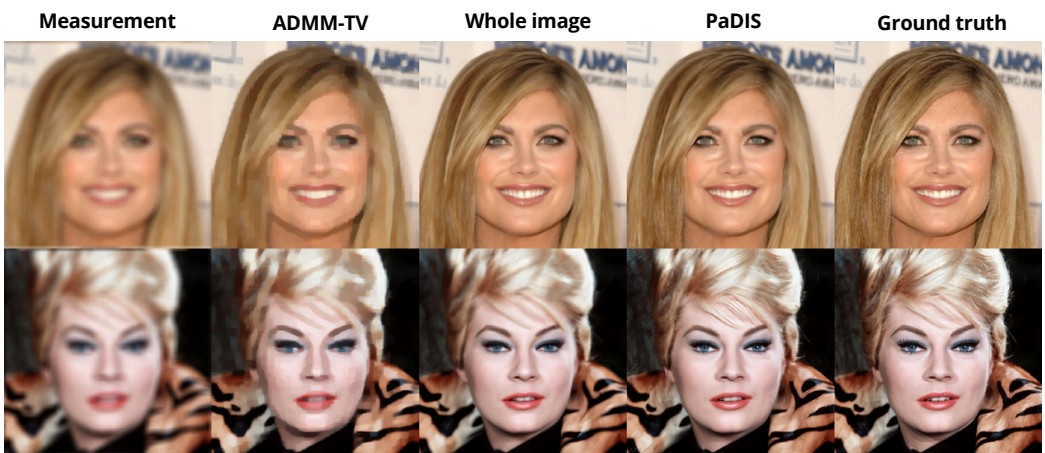

Figure 6: Results of deblurring with Gaussian noise ($\sigma = 0.01$).

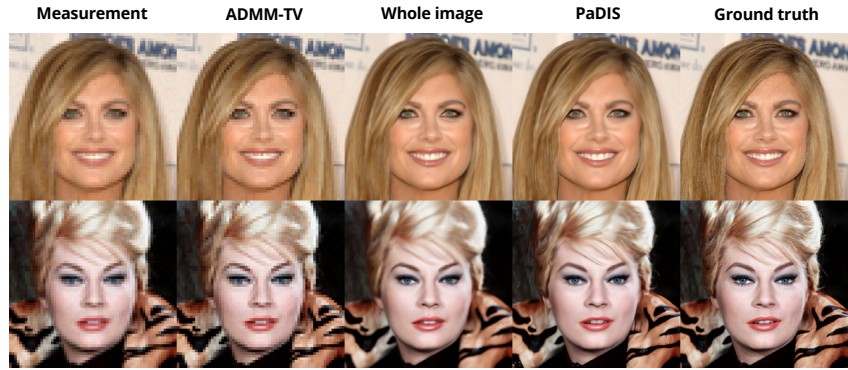

| Measurement | ADMM-TV | Whole image | PaDIS | Ground truth |

Figure 7: Results of superresolution with Gaussian noise ($\sigma = 0.01$).

Finally, since the original AAPM dataset contained CT images of resolution $512 \times 512$, we ran 60 view CT reconstruction experiments with these higher resolution images. Due to the lack of data, we did not train a whole-image model on these higher resolution images. We used a zero padding width of 64 and largest patch size of $64 \times 64$ for training. Figure 8 shows the visual results of these experiments. Hence, our proposed methods can obtain high quality reconstructions for both different inverse problems and also for different image sizes.

Table 3: Results of 60 view CT reconstruction with $512 \times 512$ images. Results are averages across all images in the test dataset. Best results are in bold.

| Method | FBP | ADMM-TV | PaDIS |
|---|---|---|---|
| PSNR ↑ | 28.38 | 29.48 | **36.93** |
| SSIM ↑ | 0.699 | 0.788 | **0.899** |

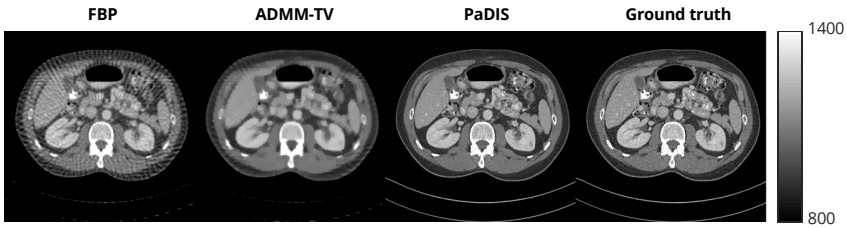

Figure 8: Results of 60 view CT reconstruction on $512 \times 512$ images using modified HU units.

## 5   Conclusion

In this work, we presented a method of using score-based diffusion models to learn image priors through solely the patches of the image, combined with suitable position encoding. Simulation results demonstrated how the method can be used to solve a variety of inverse problems. Extensive experiments showed that under conditions of limited training data, the proposed method outperforms methods involving whole image diffusion models. In the future, more work could be done on higher quality image generation using a multi-scaled resolution approach [67, 68], using acceleration methods for faster reconstruction, and higher dimensional image reconstruction. Image priors like those proposed in this paper have the potential to benefit society by reducing X-ray dose in CT scans. Generative models have the risk of inducing hallucinations and being used for disinformation.

## Acknowledgments and Disclosure of Funding

Work supported in part by a grant from the Michigan Institute for Computational Discovery and Engineering (MICDE) and a gift from KLA.

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

# A Appendix / supplemental material

This is the appendix for the paper "Learning Image Priors through Patch-based Diffusion Models for Solving Inverse Problems," presented at NeurIPS 2024.

## A.1 Additional inverse problem solving experiments

Figures A.1, A.2, A.3, and A.4 show additional inverse problem solving results.

Fig. A.1 shows additional example slices for CT reconstruction from 20 views.

Fig. A.2 shows additional example slices for CT reconstruction from 8 views.

Fig. A.3 shows additional examples of image deblurring of face images.

Fig. A.4 shows additional examples of superresolution of face images.

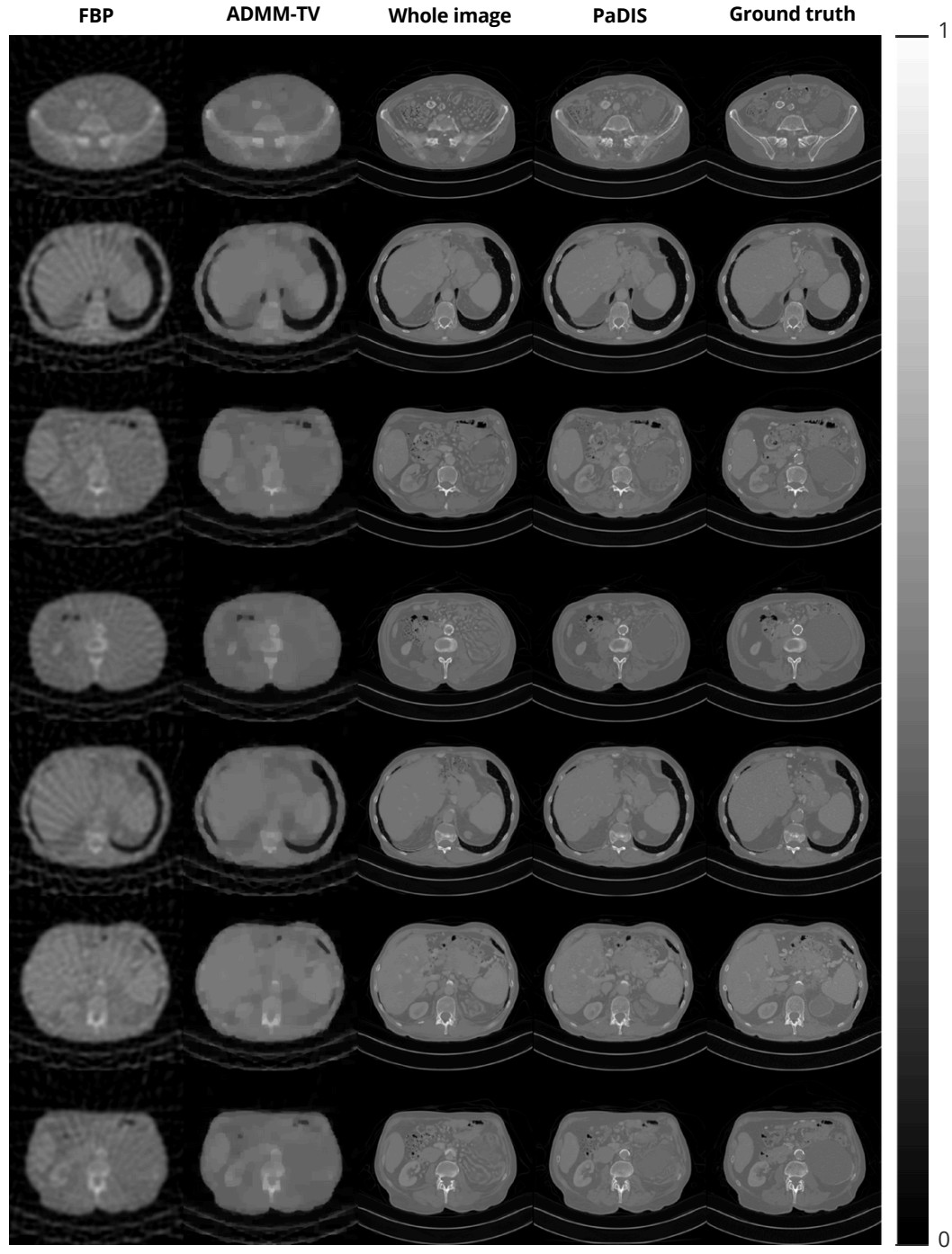

Figure A.1: Additional results of 20 view CT reconstruction for 7 different test slices.

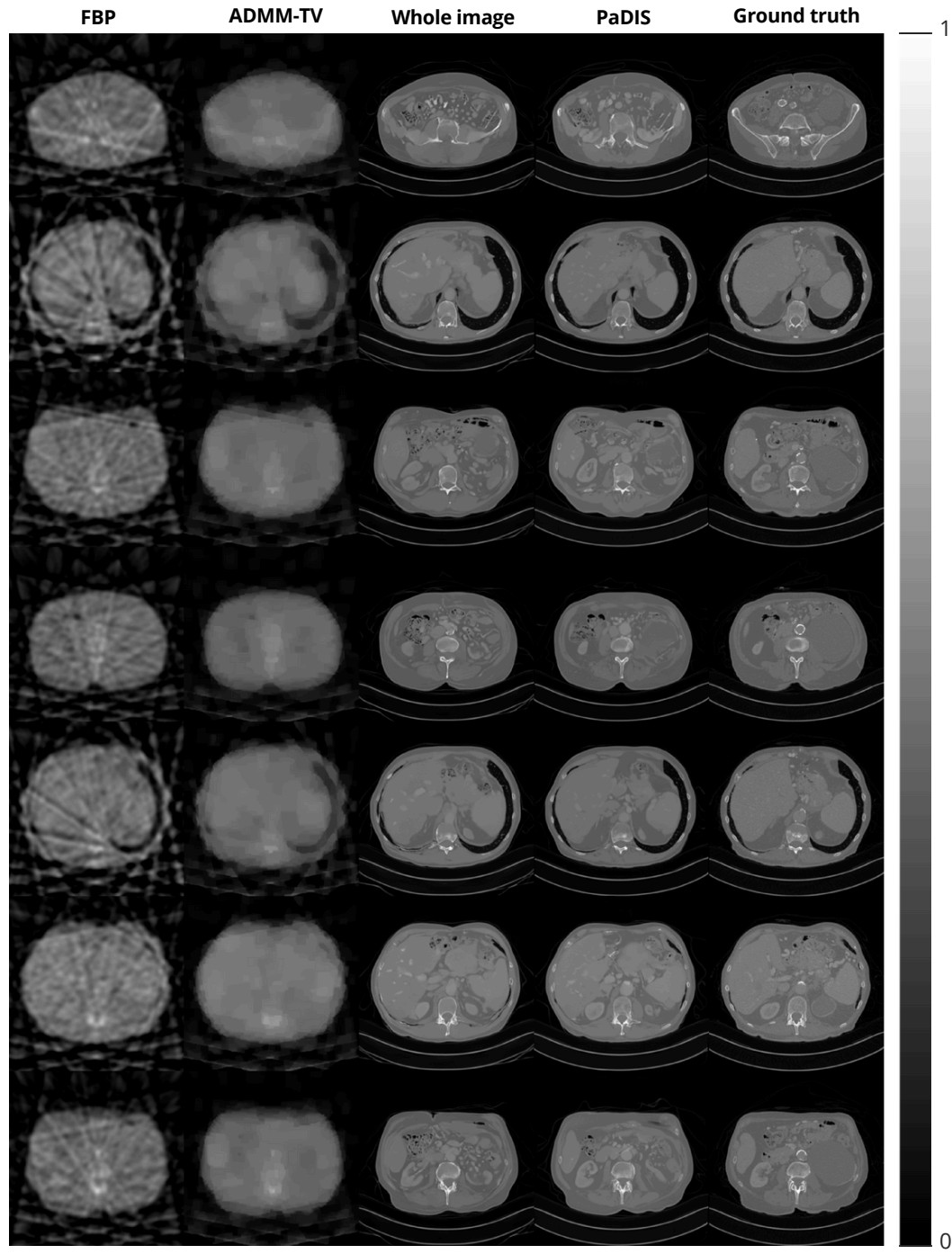

Figure A.2: Additional results of 8 view CT reconstruction for 7 different test slices.

| Measurement | ADMM-TV | Whole image | PaDIS | Ground truth |
|---|---|---|---|---|

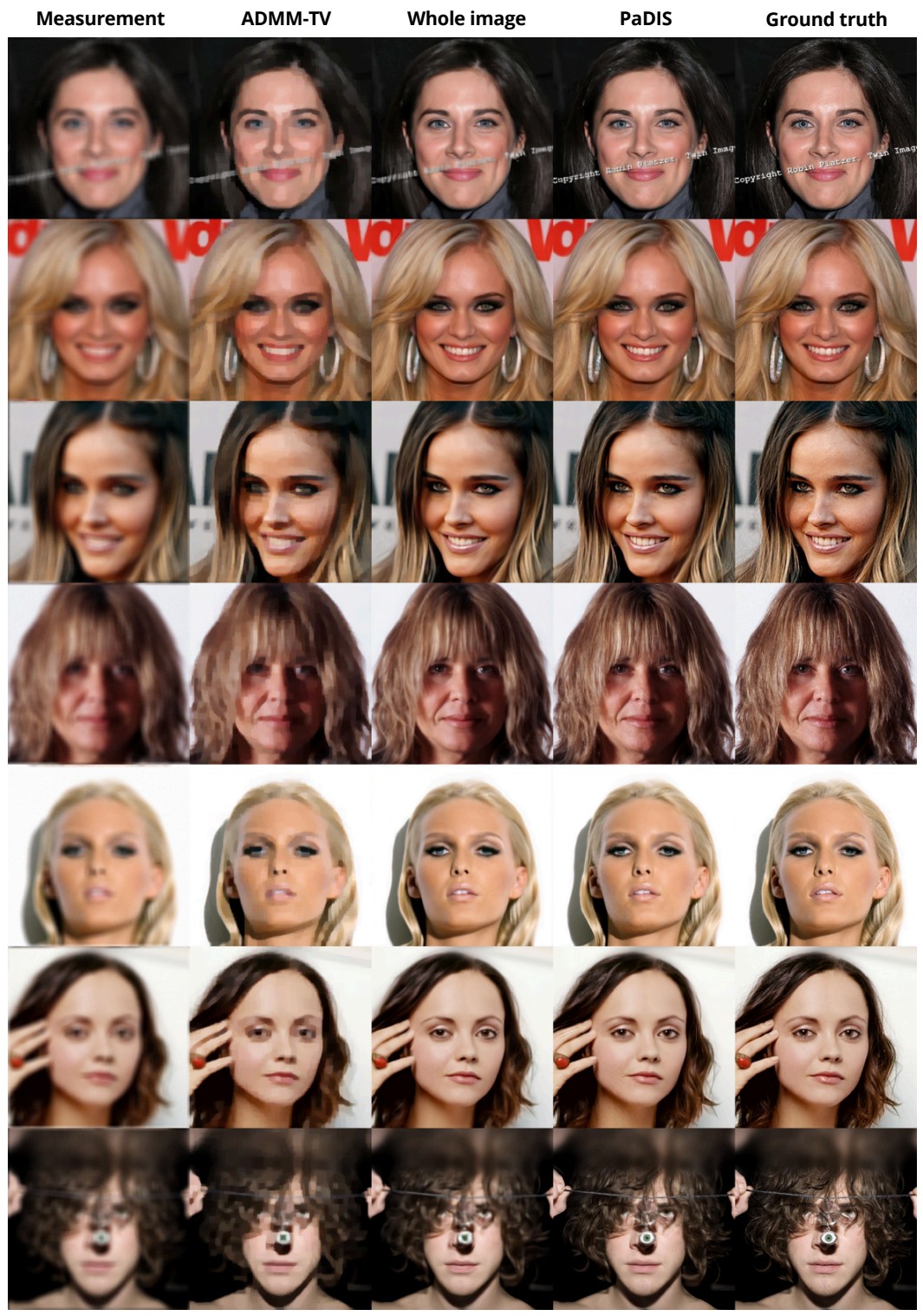

Figure A.3: Additional results of deblurring with Gaussian noise ($\sigma = 0.01$).

| Measurement | ADMM-TV | Whole image | PaDIS | Ground truth |
| --- | --- | --- | --- | --- |

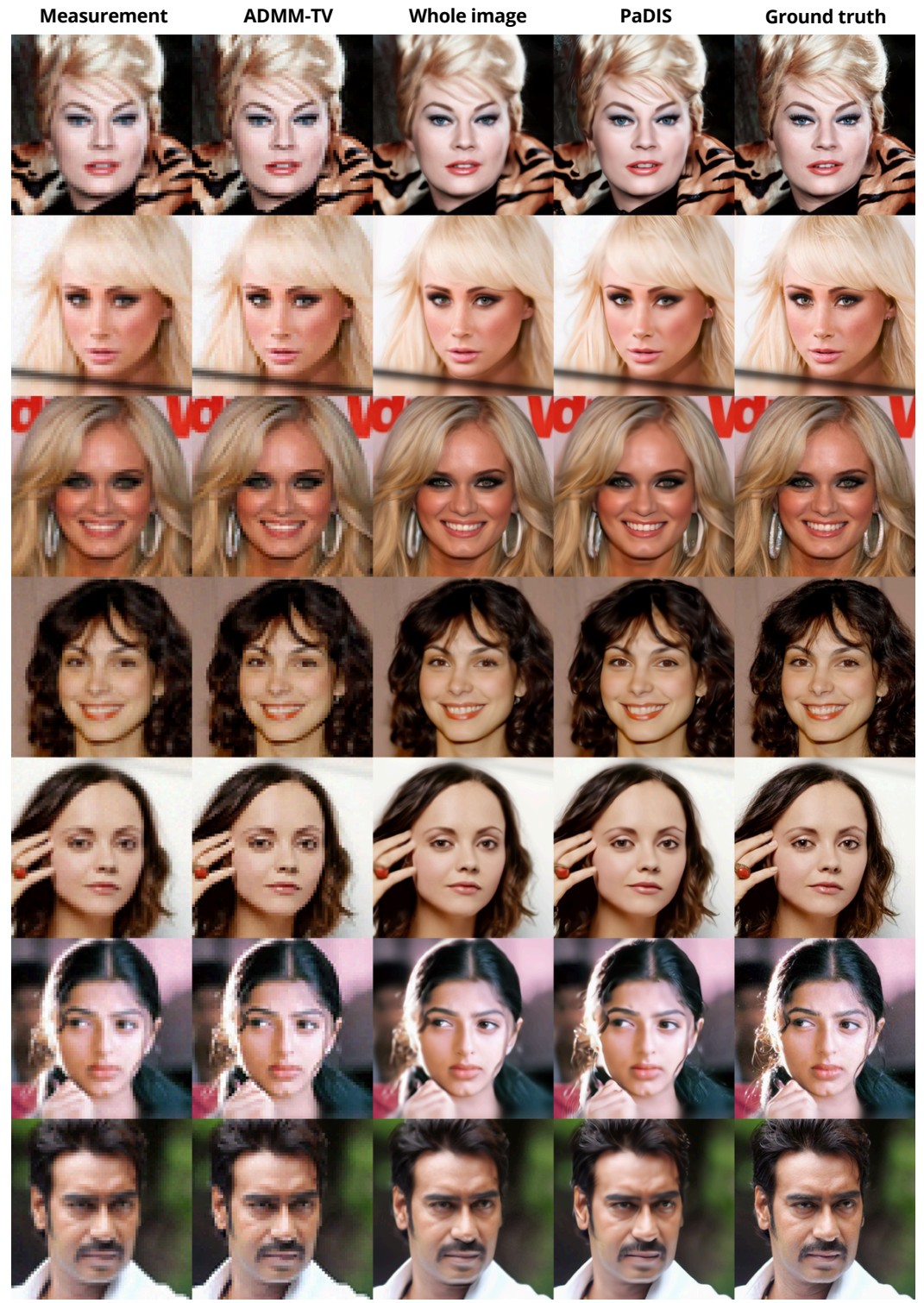

Figure A.4: Additional results of superresolution with Gaussian noise ($\sigma = 0.01$).

## A.2 Ablation studies

We performed four ablation studies to evaluate the impact of different parameters on the performance of our proposed method. Similar to Table 1, we ran the experiments on all the images in the test dataset and computed the average metric. Section A.3 shows visualizations of these studies.

**Effect of patch size.** We investigated the effect of the patch size $P$ used at reconstruction time for the 20-view CT reconstruction problem. We continued to augment the training with smaller patch sizes when possible so as to be consistent with the main experiments (patch size of 56 but also trained with patch sizes of 32 and 16), while using the same neural network architecture. Different amounts of zero padding were needed for each of the experiments per (3). App. A.4 provides the full details. At reconstruction time, the same patch size was used throughout the entire algorithm. Using a "patch size" of 256 corresponds to training a diffusion model on the whole image (without zero padding).

Table 4 shows that careful selection of the patch size is required to obtain the best results for a given training set size. If the patch size is too small, the network has trouble capturing global information across the image. Although the positional information helps in this regard, there may be some inconsistencies between patches, so the learned image prior is suboptimal although the patch priors may be learned well. At the other extreme, very large patch sizes and the whole image diffusion model require more memory to train and run. The image quality drops in this case as limited training data prevents the network from learning the patch prior well.

Table 4: Effect of patch size $P$ on CT reconstruction

| $P$ | PSNR↑ | SSIM ↑ |
|-----|-------|--------|
| 8   | 32.57 | 0.844  |
| 16  | 32.57 | 0.829  |
| 32  | 32.72 | 0.853  |
| 56  | **33.57** | **0.854** |
| 96  | 33.36 | **0.854** |
| 256 | 32.84 | 0.835  |

Table 5: Dataset size effect on CT reconstruction

| Dataset size | Patches | | Whole image | |
|--------------|---------|--------|-------------|--------|
|              | PSNR↑ | SSIM ↑ | PSNR↑ | SSIM ↑ |
| 144  | 32.28 | 0.841 | 29.12 | 0.804 |
| 288  | 32.43 | 0.837 | 31.09 | 0.829 |
| 576  | 33.03 | 0.846 | 31.81 | 0.835 |
| 1152 | 33.01 | 0.849 | 31.36 | 0.834 |
| 2304 | **33.57** | **0.854** | 32.84 | 0.835 |

**Effect of training dataset size.** A key motivation of this work is large-scale inverse problems having limited training data. To investigate the effects of using small datasets on our proposed method, compared to standard whole image models, we trained networks on random subsets of the CT dataset. Table 5 summarizes the results. Crucially, although the reconstruction quality tends to drop as the dataset size decreased for both the patch-based model and the whole image model, the drop is much more sharp and noticeable for the whole image model, particularly when the dataset is very small. This behavior is consistent with the observations of previous works where large datasets consisting of many thousands of images were used to train traditional diffusion models from scratch.

**Effect of positional encoding.** High quality image generation via patch-based models that lack positional encoding information would be impossible, as no global information about the image could be learned at all. We demonstrate that positional information is also crucial for solving inverse problems with patch-based models. We examined the results of performing CT reconstruction for trained networks *without* positional encoding as an input compared to networks *with* positional encoding. According to [69], when solving inverse problems in some settings, it can be beneficial to initialize the image with some baseline image instead of with pure noise (as is traditionally done). To allow the network that did not learn positional information to possibly use a better initialization with patches roughly in the correct positions, we also ran experiments by initializing with the baseline. Table 6 shows that in both cases, the network completely failed to learn the patch-based prior and the reconstructed results were very low quality. Hence, positional information is crucial to learning the whole image prior well.

**Sampling methods.** One benefit of our proposed method is it provides a black box image prior for the entire image that can be computed purely through neural network operations on image

patches. We demonstrate the versatility of this method by pairing a variety of different sampling and inverse problem solving algorithms with our patch-based image prior, along with comparisons with a whole-image prior. The implemented sampling methods include Langevin dynamics [1] with a gradient descent term for enforcing data fidelity step and the predictor-corrector method for solving SDEs [19]. Since we observed better stability and results with Langevin dynamics, we also combined this sampling method with nullspace methods that rely on hard constraints [7] and DPS [5]. To use the same neural network checkpoint across these implementations, we used the variance exploding SDE [2] method as the backbone for both training and reconstruction. DPS [5] and DDNM [7] were originally implemented with networks trained under the VP-SDE framework; here, we implemented those methods with the VE-SDE framework. Table 7 shows that generally, VE-DPS performed the best and that the patch-based method consistently outperformed the whole image method. However, the patch-based method still obtained reasonable results for all the implemented methods, showing that the learned image prior is indeed flexible enough to be paired with a variety of sampling algorithms. App. A.4 provides more details about the implemented algorithms.

Table 6: Positional encoding effect for CT reconstruction

|  | PSNR↑ | SSIM ↑ |
|---|---|---|
| no position enc. | 23.25 | 0.459 |
| no position+init | 24.51 | 0.518 |
| with position | **33.57** | **0.854** |

Table 7: Dataset size effect on CT reconstruction

| Method | Patch-based | | Whole image | |
|---|---|---|---|---|
| Metric | PSNR↑ | SSIM ↑ | PSNR↑ | SSIM ↑ |
| Langevin dynamics | 33.03 | 0.846 | 30.92 | 0.813 |
| Predictor-corrector | 32.35 | 0.820 | 18.95 | 0.149 |
| VE-DDNM | 31.98 | **0.861** | 29.49 | 0.830 |
| VE-DPS | **33.57** | 0.854 | 32.84 | 0.835 |

### A.3 Ablation study images

Figure A.5 shows the results of applying PaDIS to two example test images with different patch sizes. The main results, i.e., those shown in Table 1, used $P = 56$. For some of the other patch sizes, some artifacts can be seen in the images. Namely, the smooth parts of the image become riddled with "fake" features for small patch sizes and some of the sharp features become more blurred. The fake features in the right half of the image in the top row are especially apparent when applying the whole-image model. The runtime for different patch sizes were fairly similar, with $P = 8$ taking notably longer than the others due to the large number of patches required. The image size for these experiments was small enough so that the score function of all the patches could be computed in parallel; however, for larger scale problems such as high resolution 2D images or 3D images, large patch sizes become infeasible due to memory constraints.

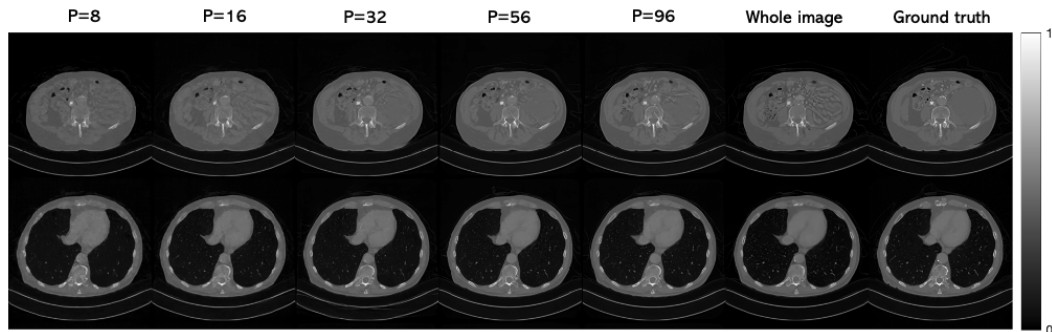

Figure A.5: Results of PaDIS for 20 view CT reconstruction with different sized patches.

Figure A.6 shows the results of applying our proposed method and the whole image diffusion model to 20-view CT reconstruction for varying sizes of the training dataset. The image quality for PaDIS remains visually consistent as the size of the training dataset shrinks, as each image contains thousands of patches which helps avoid overfitting and memorization. However, the drop in quality for the whole image model is much more visible: in particular, the sharp features of the image are lost and the image becomes blurry. Hence, for applications where data is even more limited, such as medical imaging, our method can potentially have a greater benefit.

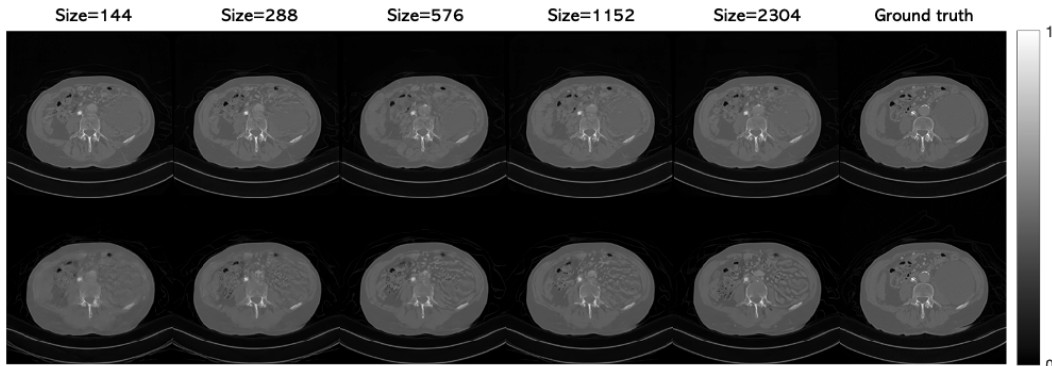

Figure A.6: Results for 20 view CT reconstruction with different dataset sizes. Top row shows recon performed by PaDIS; bottom row shows recon performed with the whole-image model.

Figure A.7 demonstrates the importance of adding positional encoding information into the patch-based network on two different images. When positional information is not included, the network simply learns a mixture of all patches, resulting in a very blurry image with many artifacts resulting from the data fidelity term. Even when a better initialization of the image is provided, the same blurriness remains.

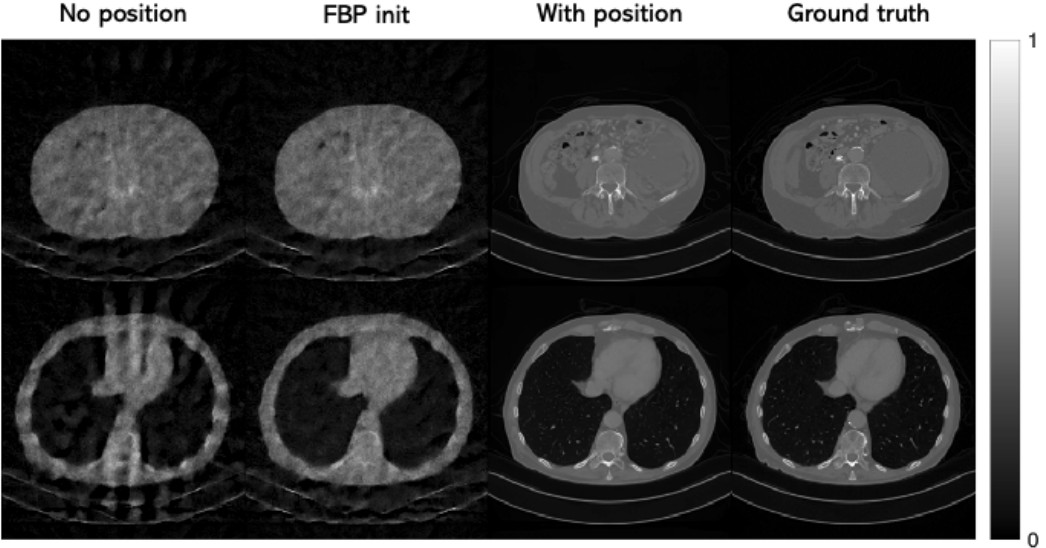

Figure A.7: Results of PaDIS for 20 view CT reconstruction for different positional encoding methods.

Figure A.8 shows the results of using our proposed method compared with the whole-image diffusion model with different sampling and inverse problem solving algorithms. The predictor-corrector algorithm fails completely when using the whole-image model, indicating that this model could not be well-trained in this limited data setting. Quantitatively, DPS performs the best for PaDIS; visually, all of the methods obtain reasonable results, although some more minor artifacts are present in the first four methods. Nevertheless, this shows that the patch-based prior is flexible and can be used with a variety of existing algorithms.

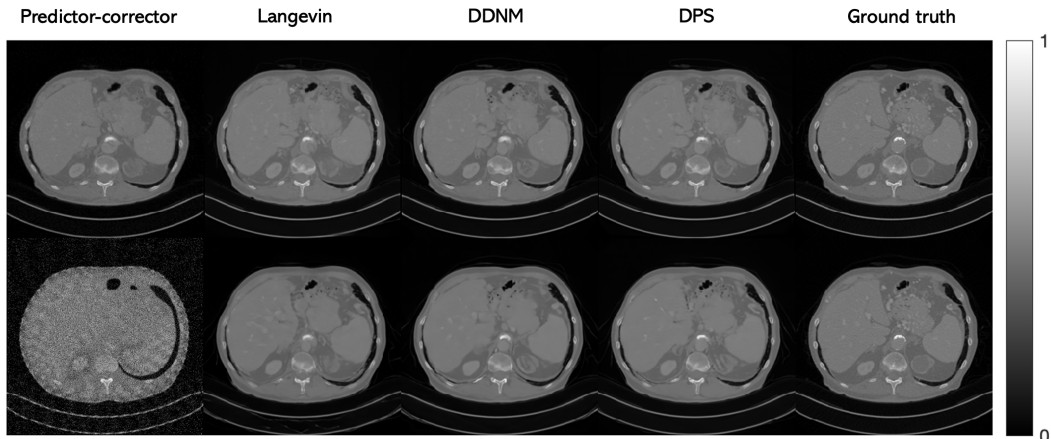

Figure A.8: Results of PaDIS for 20 view CT reconstruction using different sampling and inverse problem solving algorithms. Top row is with PaDIS and bottom row is with the whole-image model.

## A.4  Experiment parameters

We trained the patch-based networks and whole-image networks following [14]. Since images were scaled between 0 and 1, we chose a maximum noise level of $\sigma = 40$ and a minimum noise level of $\sigma = 0.002$. We used the same UNet architecture for all the patch-based networks consisting of a base channel multiplier size of 128 and 1, 2, 2, and 2 channels per resolution for the four layers. We also used dropout connections with a probability of 0.05 and exponential moving average for weight decay with a half life of 500K patches to avoid overfitting. Finally, the learning rate was chosen to be $2 \cdot 10^{-4}$ and the batch size for the main patch size was 128, although batch sizes of 256 and 512 were used for the two smaller patch sizes. The entire model had around 60 million weights. For the whole image model, we kept all the parameters the same, but increased the number of channels per resolution in the fourth layer to 4 so that the model had around 110 million weights. The batch size in this case was 8.

For image generation and solving inverse problems, we used a geometrically spaced descending noise level that was fine tuned to optimize the performance for each type of problem. We used the same set of parameters for the patch-based model and whole image model, as follows:

- CT with 20 views: $\sigma_{\max} = 10, \sigma_{\min} = 0.002$
- CT with 8 views: $\sigma_{\max} = 10, \sigma_{\min} = 0.003$
- Deblurring: $\sigma_{\max} = 40, \sigma_{\min} = 0.005$
- Superresolution: $\sigma_{\max} = 40, \sigma_{\min} = 0.01$.

The ADMM-TV method for linear inverse problems consists of solving the optimization problem

$$\operatorname{argmax}_{\boldsymbol{x}} \frac{1}{2}\|\boldsymbol{y} - A\boldsymbol{x}\|_2^2 + \lambda \operatorname{TV}(\boldsymbol{x}), \tag{A.1}$$

where $\operatorname{TV}(\boldsymbol{x})$ represents the L1 norm total variation of $\boldsymbol{x}$, and the problem is solved with the alternating direction method of multipliers. For CT reconstruction, deblurring, and superresolution, we chose $\lambda$ to be $0.001, 0.002$, and $0.006$ respectively.

**Ablation study details.** For each patch size, we trained with the main patch size along with smaller patches whenever possible. However, since we did not modify the network architecture, and the architecture consists of downsampling the image 3 times by a factor of 2, it was necessary for the input dimension to be a multiple of 8. Furthermore, we followed a patch scheduling method similar to that of the main experiments unless otherwise noted. Finally, to avoid excessive zero padding, for larger patch sizes, we used patch sizes that were smaller than the next power of 2 such that the main image could still be fully covered by the same number of patches. The details are as follows.

- $P = 8$: This was trained only with this patch size as no smaller sizes could be used.
- $P = 16$: Trained with patch sizes of 8 and 16 with probabilities of 0.3 and 0.7 respectively.
- $P = 32$: Trained with patch sizes of 8, 16, and 32 with probabilities of 0.2, 0.3, and 0.5 respectively.
- $P = 56$: Trained with patch sizes of 16, 32, and 56 with probabilities of 0.2, 0.3, and 0.5 respectively. Zero padding width was set to $5 \cdot 56 - 256 = 24$.
- $P = 96$: Trained with patch sizes of 32, 64, and 96 with probabilities of 0.2, 0.3, and 0.5 respectively. Zero padding width was set to $3 \cdot 96 - 256 = 32$.

## A.5 Comparison algorithms

This section provides pseudocode for the implemented alternative sampling algorithms whose results are shown in Table 7. Here, for brevity, we show the versions using the whole-image diffusion model; the versions with our proposed method are readily implemented by computing $\boldsymbol{s} = \boldsymbol{s}(\boldsymbol{x}, \sigma_i)$ through the procedure illustrated in Alg. 1.

---
**Algorithm A.1** Image Recon via Langevin Dynamics

---
**Require:** $\sigma_1 < \sigma_2 < \ldots < \sigma_T, \epsilon > 0, \zeta_i > 0, \boldsymbol{y}$
  Initialize $\boldsymbol{x} \sim \mathcal{N}(0, \sigma_T^2 \boldsymbol{I})$
  **for** $i = T : 1$ **do**
    Sample $z \sim \mathcal{N}(0, \sigma_i^2 \boldsymbol{I})$
    Set $\alpha_i = \epsilon \cdot \sigma_i^2$
    Apply neural network to get $D = D_\theta(\boldsymbol{x}, \sigma_i)$
    Set $\boldsymbol{s} = (D - \boldsymbol{x})/\sigma_i^2$
    Set $\boldsymbol{x}$ to $\boldsymbol{x} + \zeta_i \mathcal{A}^T(y - \mathcal{A}(\boldsymbol{x}))$
    Set $\boldsymbol{x}$ to $\boldsymbol{x} + \frac{\alpha_i}{2}\boldsymbol{s} + \sqrt{\alpha_i}\boldsymbol{z}$
  **end for**

---
Return $\boldsymbol{x}$.

---

---
**Algorithm A.2** Image Recon via Predictor-Corrector Sampling

---
**Require:** $\sigma_1 < \sigma_2 < \ldots < \sigma_T, \epsilon > 0, \zeta_i > 0, r, \boldsymbol{y}$
  Initialize $\boldsymbol{x} \sim \mathcal{N}(0, \sigma_T^2 \boldsymbol{I})$
  **for** $i = T : 1$ **do**
    Set $\boldsymbol{x}$ to $\boldsymbol{x} + (\sigma_{i+1}^2 - \sigma_i^2)\boldsymbol{s}_\theta(x, \sigma_{i+1})$
    Set $\boldsymbol{x}$ to $\boldsymbol{x} + \zeta_i \mathcal{A}^T(y - \mathcal{A}(\boldsymbol{x}))$
    Sample $\boldsymbol{z} \sim \mathcal{N}(0, \boldsymbol{I})$
    Set $\boldsymbol{x}$ to $\boldsymbol{x} + \sqrt{\sigma_{i+1}^2 - \sigma_i^2}\boldsymbol{z}$
    Sample $\boldsymbol{z} \sim \mathcal{N}(0, \boldsymbol{I})$
    Set $\epsilon_i = 2r\frac{\|z\|_2}{\|\boldsymbol{s}_\theta(\boldsymbol{x}, \sigma_i)\|_2}$
    Set $\boldsymbol{s} = \boldsymbol{s}_\theta(\boldsymbol{x}, \sigma_i)$
    Set $\boldsymbol{x}$ to $\boldsymbol{x} + \epsilon_i \boldsymbol{s} + \sqrt{2\epsilon_i}\boldsymbol{z}$
    Set $\boldsymbol{x}$ to $\boldsymbol{x} + \zeta_i \mathcal{A}^T(y - \mathcal{A}(\boldsymbol{x}))$
  **end for**

---
Return $\boldsymbol{x}$.

---

In all cases, we used the same noise schedule as the main 20 view CT reconstruction experiment. For Langevin dynamics and DDNM, we set $\epsilon = 1$; the final results were not sensitive with respect to this

**Algorithm A.3** DDNM

---

**Require:** $\sigma_1 < \sigma_2 < \ldots < \sigma_T, \epsilon > 0, \zeta_i > 0, \boldsymbol{y}$
    Initialize $\boldsymbol{x} \sim \mathcal{N}(0, \sigma_T^2 \boldsymbol{I})$
    **for** $i = T : 1$ **do**
        Sample $z \sim \mathcal{N}(0, \sigma_i^2 \boldsymbol{I})$
        Set $\alpha_i = \epsilon \cdot \sigma_i^2$
        Apply neural network to get $D = D_\theta(\boldsymbol{x}, \sigma_i)$
        Set $D = \mathcal{A}^\dagger \boldsymbol{y} + D - \mathcal{A}^\dagger \mathcal{A}(D)$
        Set $\boldsymbol{s} = (D - \boldsymbol{x})/\sigma_i^2$
        Set $\boldsymbol{x}$ to $\boldsymbol{x} + \frac{\alpha_i}{2}\boldsymbol{s} + \sqrt{\alpha_i}\boldsymbol{z}$
    **end for**
Return $\boldsymbol{x}$.

---

parameter. For Langevin dynamics and predictor-corrector sampling, we took $\zeta_i = 0.3/\|\boldsymbol{y} - \mathcal{A}(\boldsymbol{x})\|_2$, similar to the step size selection of DPS. Following the work of [19], we chose $r = 0.16$ for PC-sampling. The same parameters were used for the patch-based and whole image methods.

Table 8 shows the average runtimes of each of the implemented methods when averaged across the test dataset for 20 view CT reconstruction.

Table 8: Average runtimes of different methods across images in the test dataset for 20 view CT recon.

| Method | Runtime (s) $\downarrow$ |
|---|---|
| Baseline | 0.1 |
| ADMM-TV | 1 |
| PnP-ADMM | 8 |
| PnP-RED | 22 |
| Whole image diffusion | 172 |
| Langevin dynamics | 98 |
| Predictor-corrector | 189 |
| VE-DDNM | 105 |
| PaDIS (VE-DPS) | 195 |

## A.6 Markov random field interpretation

Markov random fields (MRF) are a tool used to represent certain image distributions and are particularly applicable to patch-based diffusion models. Describing the connection between MRF and this work requires some notation: let $\boldsymbol{x} = \{x_s : s \in \mathcal{S}\}$ denote the random field, where the index $\mathcal{S}$ denotes the sites. The neighborhood system $\mathcal{N}$ is defined as $\mathcal{N} = \{\mathcal{N}_s : s \in S\}$. A model for $\boldsymbol{x} \in \mathcal{X}$ is a MRF on $\mathcal{S}$ with respect to the neighborhood system $\mathcal{N}$ if

$$p(x_s|x_{\mathcal{S}-\{s\}}) = p(x_s|x_{\mathcal{N}_s}), \; \forall \boldsymbol{x} \in \mathcal{X}, \; \forall s \in \mathcal{S}. \tag{A.2}$$

Therefore, the distribution of each site (normally chosen to be a pixel) conditioned on the rest of the pixels depends only on the neighboring pixels.

By the Hammersley-Clifford theorem [70], such a MRF satisfying $p(\boldsymbol{x}) > 0$ everywhere can also be rewritten as $p(\boldsymbol{x}) = \frac{1}{Z}e^{-U(\boldsymbol{x})}$, where $Z$ is a normalizing constant. In this case $U(\boldsymbol{x})$ is called the energy function and has the form $U(\boldsymbol{x}) = \sum_{c \in \mathcal{C}} V_c(\boldsymbol{x})$, which is a sum of clique potentials $V_c(\boldsymbol{x})$ over all all possible cliques. Thus, the score function for a MRF model is:

$$\boldsymbol{s}(\boldsymbol{x}) = \nabla \log p(\boldsymbol{x}) = -\nabla U(\boldsymbol{x}) = -\sum_c \nabla V_c(\boldsymbol{x}). \tag{A.3}$$

If we let the neighborhood system be the patches of the image, then $V_c$ corresponds to the clique potential for the $c$th patch of an image, and $-\nabla V_c(\boldsymbol{x})$ denotes the score function of that patch. Denoting by $\boldsymbol{G}_c$ the wide binary matrix that extracts the pixels corresponding to the $c$th patch from the whole image, we define $V_c(\boldsymbol{x}) = V(\boldsymbol{G}_c\boldsymbol{x}, c^*)$, where $c^*$ denotes the positional encoding method used for the $c$th patch, and now we simply have one (patch) clique function $V$. Finally, the overall score function under this model becomes

$$\boldsymbol{s}(\boldsymbol{x}) = \sum_c \boldsymbol{G}_c' \boldsymbol{s}_V(\boldsymbol{G}_c\boldsymbol{x}, c^*), \tag{A.4}$$

where $\boldsymbol{s}_V(\boldsymbol{v}, c^*) \triangleq -\nabla_{\boldsymbol{v}} V(\boldsymbol{v}, c^*)$ is the shared score function of each of the patches with a positional encoding input.

In this work, we approximate $\boldsymbol{s}_V$ with a neural network parameterized by $\theta$, and we use denoising score matching to train the network via the loss function

$$L(\theta) = \mathbb{E}_{t \sim \mathcal{U}(0,T)} \mathbb{E}_{\boldsymbol{x} \sim p(\boldsymbol{x})} \mathbb{E}_{\boldsymbol{\epsilon} \sim \mathcal{N}(0,\sigma_t^2 \boldsymbol{I})} \|\boldsymbol{s}_\theta(\boldsymbol{x} + \boldsymbol{\epsilon}, \sigma_t) + \boldsymbol{\epsilon}/\sigma_t^2\|_2^2 \tag{A.5}$$

$$= \mathbb{E}_{t \sim \mathcal{U}(0,T)} \mathbb{E}_{\boldsymbol{x} \sim p(\boldsymbol{x})} \mathbb{E}_{\boldsymbol{\epsilon} \sim \mathcal{N}(0,\sigma_t^2 \boldsymbol{I})} \left\| \sum_c \boldsymbol{G}_c' \boldsymbol{s}_V(\boldsymbol{G}_c\boldsymbol{x}, c^*; \theta) + \boldsymbol{\epsilon}/\sigma_t^2 \right\|_2^2. \tag{A.6}$$

This derivation makes no assumptions on the patches; in particular, this method to train the score function would still hold if the patches overlapped for each iteration. However, such overlap would make it costly to train the network, as the loss function would need to be propagated through the sum over all patches every training iteration. We circumvent this problem by using non-overlapping patches (within a given reconstruction iteration; i.e., our approach only uses patches that "overlap" only across different iterations). For non-overlapping patches, the sum can be rewritten as follows:

$$L(\theta) = \mathbb{E}_{t \sim \mathcal{U}(0,T)} \mathbb{E}_{\boldsymbol{x} \sim p(\boldsymbol{x})} \mathbb{E}_{\boldsymbol{\epsilon} \sim \mathcal{N}(0,\sigma_t^2 \boldsymbol{I})} \left\| \sum_c \boldsymbol{G}_c' \boldsymbol{s}_V(\boldsymbol{G}_c\boldsymbol{x}, c^*; \theta) + \boldsymbol{G}_c' \boldsymbol{G}_c \boldsymbol{\epsilon}/\sigma_t^2 \right\|_2^2 \tag{A.7}$$

$$= \mathbb{E}_{t \sim \mathcal{U}(0,T)} \mathbb{E}_{\boldsymbol{x} \sim p(\boldsymbol{x})} \mathbb{E}_{\boldsymbol{\epsilon} \sim \mathcal{N}(0,\sigma_t^2 \boldsymbol{I})} \mathbb{E}_{\text{random } c} \|\boldsymbol{s}_V(\boldsymbol{G}_c\boldsymbol{x}, c^*; \theta) + \boldsymbol{G}_c\boldsymbol{\epsilon}/\sigma_t^2\|_2^2. \tag{A.8}$$

This loss function is much easier to compute, as we can now randomly select patches and perform denoising score matching on the individual patches, as opposed to considering the entire image at once, and is equivalent to (A.5).

## A.7 Acceleration methods

Although diffusion models are capable of generating high quality images, the iterative generative process typically requiring around 1000 neural function evaluations (NFEs) [2, 3] is a major disadvantage. In recent years, significant work has been done to improve sampling speed of diffusion models [14, 30, 31]. To reuse the same trained network, one can first derive an algebraic relationship between the score function $s(x, \sigma)$ (readily computed using the denoiser network $D(x, \sigma)$ trained via (5)) and the residual function $\epsilon(x, t)$ which is approximated with a neural network in papers such as [3]. The score matching network $s_\theta(x, \theta)$ learns to map $x + \sigma\epsilon$ to $x$, whereas the residual network learns to map $x_t = \sqrt{\alpha_t}x_0 + \sqrt{1 - \alpha_t}\epsilon$ to $\epsilon$ where $\epsilon \sim \mathcal{N}(0, I)$. Hence, we may input $\frac{x_t}{\sqrt{\alpha_t}} = x_0 + \frac{\sqrt{1-\alpha_t}}{\sqrt{\alpha_t}}\epsilon$ as the noisy image into the denoising score matching network so that the correct output becomes $x_0$. Then the corresponding noise level is $\sigma_t = \frac{\sqrt{1-\alpha_t}}{\sqrt{\alpha_t}}$. Finally, the outputs of the network must be scaled via $s_\theta(x) = -\epsilon(x)/\sigma$. Thus, by using this transformation, the network trained via (5) may also be applied to sampling algorithms requiring $\epsilon_\theta$.

Using these ideas, we implemented the EDM sampler, a second-order solver for SDEs, according to [14], which can produce high fidelity images in 18 iterations (equating to 36 NFEs as each iteration requires two NFEs). We also implemented DDIM [30] using 50 sampling steps. Figure A.9 shows the results of using these methods with the proposed patch-based prior along with Langevin dynamics with 300 NFEs. The EDM sampler produces images that have clear boundary artifacts and the images from the DDIM method also have some discontinuous parts. This behavior is due to the stochastic method of computing the patch-based prior: according to Algorithm 1, we randomly choose integers $i$ and $j$ with which to partition the zero padded image and compute the score function according to this partition. Hence, accelerated sampling algorithms that attempt to remove large amounts of noise at each step tend to fare worse at removing boundary artifacts. This limitation of our method makes it difficult to run accelerated sampling algorithms, so is a direction for future research.

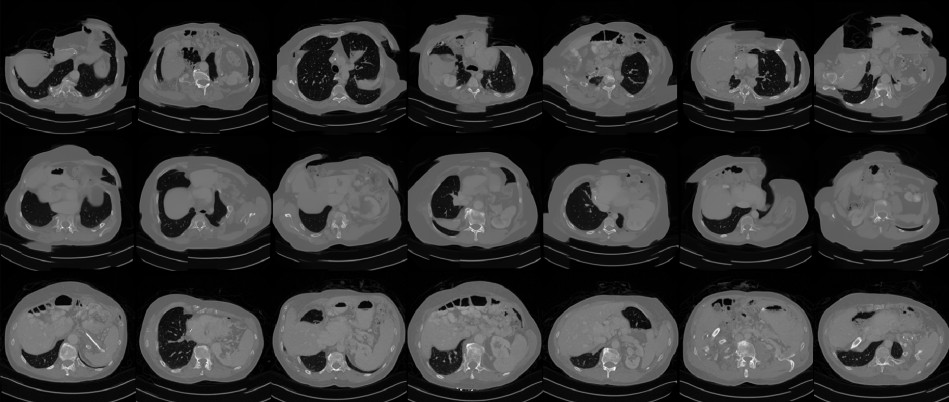

Figure A.9: Generation of CT images with various acceleration algorithms. Top row shows generation with the EDM sampler [14], middle row uses DDIM [30], bottom row is our proposed method.

## A.8    Additional figures

Figure A.10 shows the visual results of the extra inverse problems in Table 2.

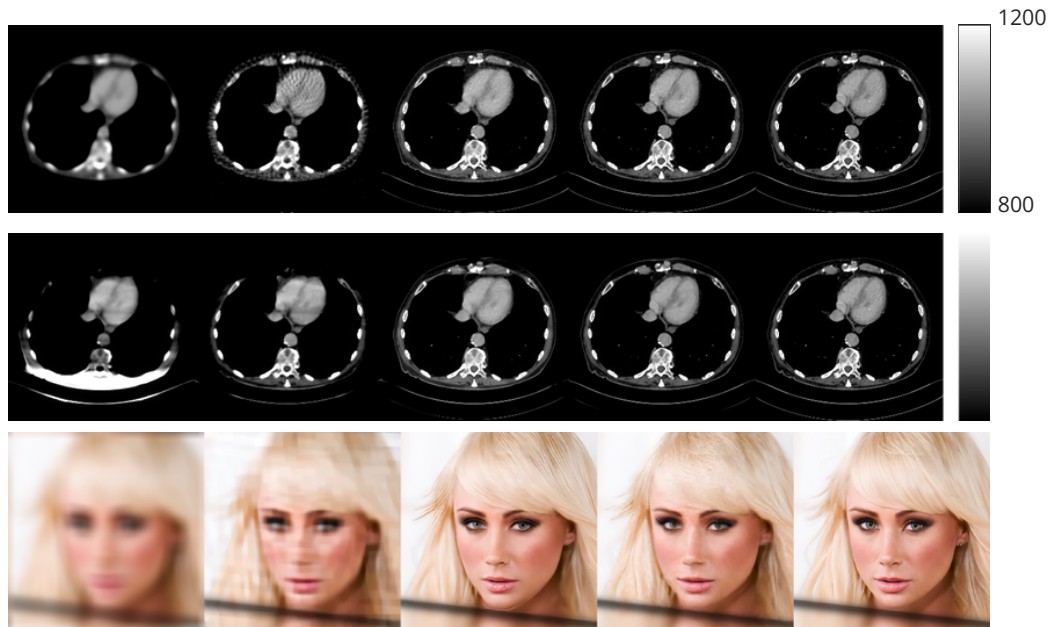

Figure A.10: Results of extra inverse problem experiments. From top to bottom: 60 view CT, fan beam CT, heavy deblurring. From left to right: baseline, ADMM-TV, whole image diffusion, PaDIS, ground truth.

To further explore the different methods of assembling patches to form the whole image, we looked at unconditionally generated CT images according to the methods of [23] and [66]. Unlike with our proposed method, for these two methods, the patch locations are fixed throughout all of the timesteps. Furthermore, overlapping patches must be used to avoid boundary artifacts. The main difference is the way in which the methods handle the overlapping pixels: [66] *overrides* the overlapped areas with the new patch update while [23] *averages* over the overlapped area. We use the same network checkpoint trained on CT images as the main experiments and a patch size of 56 with overlap of 8 for the experiments. Figure A.11 shows the generated images: while both methods are able to avoid boundary artifacts, the overall structure of the generations are of lower quality than the images generated by our proposed method (see Table 4). This suggests that our proposed method most effectively combines the patch priors to form a prior for the entire image.

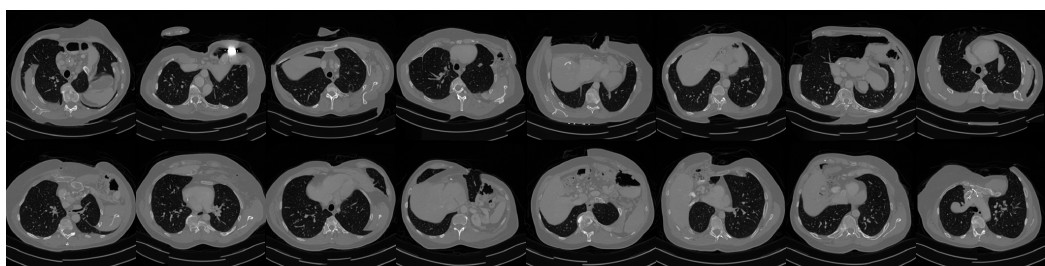

Figure A.11: Unconditionally generated CT images using patch stitching [66] for the top row and patch averaging [23] for the bottom row. Compare this to the images generated by our proposed method in Figure 4.

Figures A.12 and A.13 show the PSNR of each individual image in the test dataset when using the whole-image model versus our proposed method. The plots show that our method exhibited reasonably consistent performance improvements over the test dataset.

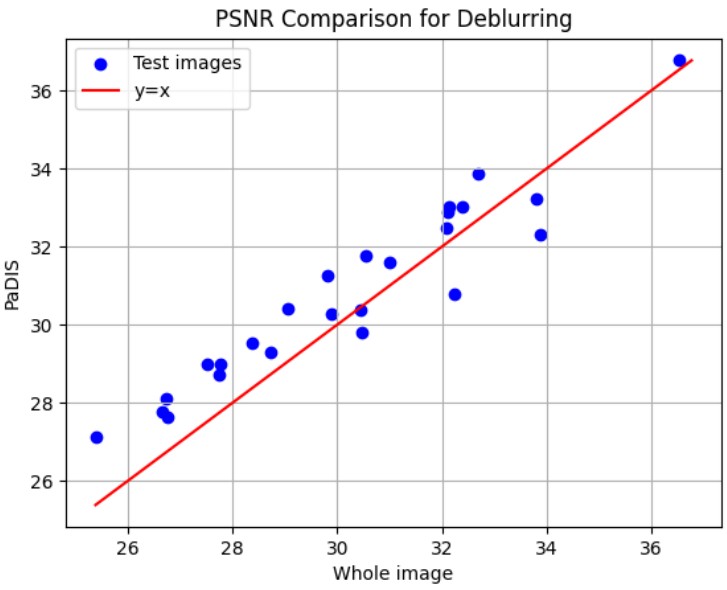

Figure A.12: Comparison between PSNR of deblurring between whole-image model and proposed method for each image in the test dataset.

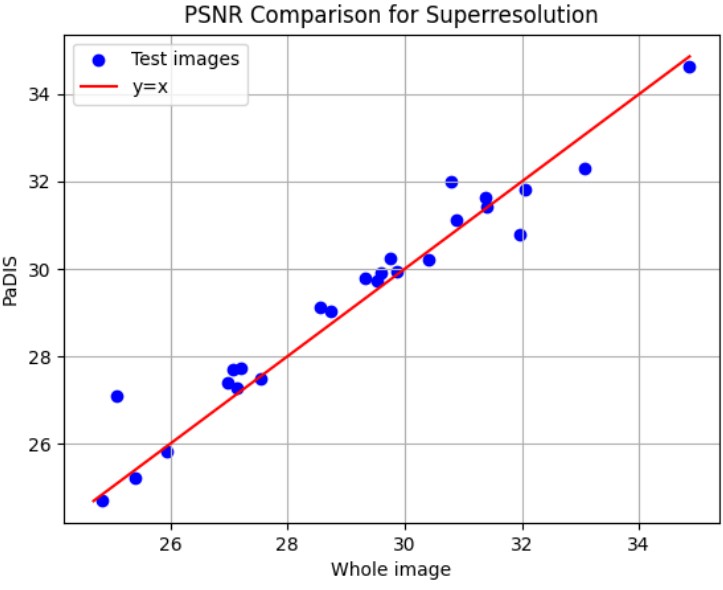

Figure A.13: Comparison between PSNR of superresolution between whole-image model and proposed method for each image in the test dataset.

