# OpenReview forum: "Learning Image Priors Through Patch-Based Diffusion Models for Solving Inverse Problems"
_NeurIPS.cc/2024/Conference — NeurIPS 2024 poster_

### Official Review · Reviewer_wfxx · 2024-06-25

**Soundness:** 3
**Presentation:** 3
**Contribution:** 3
**Rating:** 6
**Confidence:** 4

**Summary:**

This work introduces a patch-based diffusion modeling approach to efficiently learn image priors that can be used to solve inverse problems. Particularly the model maintains memory and data efficiency due to the patch-based operating scheme. Experiments are demonstrated in both natural and medical image domains to solve various inverse problems (e.g., CT reconstruction, deblurring, etc.) based on priors learned via patches.

**Strengths:**

- Paper presents a very interesting and novel approach to learn image priors with patch-based diffusion models, and empirically demonstrates rigorous results.

**Weaknesses:**

- Some experimental clarifications are necessary, and writing & quality of figures can be improved.

**Questions:**

- Details regarding how the evaluation metrics PSNR and SSIM are calculated are missing (i.e., in RGB domain, or via the luminance in YCbCr domain)?

- The authors' justification on using non-overlapping patches during training is somewhat not clear to me. What exactly is the negative impact of using overlapping patches during training this model?

- In Figure 4, results obtained with [16] are surprisingly bad. Is the implementation modified, or did the authors implement themselves? Is it the best case scenario for this method's performance? Also on a different note, did the authors perhaps try using the patch-based diffusion approach from [23] in an unsupervised manner in this simulation?

- Considering Algorithm 1's sampling loop, looks like the sampling process goes through all denoising steps without skips (sampling takes T steps)? In that case, perhaps inference time comparisons should also be demonstrated comparatively with other methods.

- Visual results (e.g., in Figures 5, 6, 7) should be blended in the compiled PDF not as a PNG/JPEG, but as a high resolution image as a e.g., PDF (more professional looking figure quality needed). Fine differences/details that one should see when zoomed-in can disappear if figures are blended in the paper using a PNG/JPEG type of format, which loses its meaning.

**Limitations:**

Sufficiently addressed.

---

> ### Author Rebuttal · Authors · 2024-08-06
>
> We thank the reviewer for their insightful comments.
>
> **Comment: Details regarding calculations of PSNR and SSIM are missing**
>
> The PSNR and SSIM of RGB images are calculated in the RGB domain. Data preprocessing consisting of dividing all the RGB values by 255 was done first, so all the reconstructed images have values between 0 and 1. The PSNR and SSIM values were then computed from these images.
>
> **Comment: Justification for using non-overlapping patches is unclear**
>
> Using overlapping patches during training would significantly increase the computational cost. Section A.6 details the theory behind which the original score matching process (that must be done on the entire image), used to train the network, can be reduced to score matching on individual patches. In particular, in going from equation (A.6) to equations (A.7) and (A.8), the assumption that the patches do not overlap is made to bring the sum out of the norm. If overlapping patches were to be used, it would be necessary to backpropagate through all the terms of the sum during the training. On the other hand, when nonoverlapping patches are used, we can perform score matching on individual patches and only the loss of these individual patches needs to be backpropagated through the network.
>
> **Comment: Results obtained by [16] are surprisingly bad**
>
> In Figure 4, for [16], we used the code shared by the paper’s authors. However, there is a key difference between the figures we generated and the figures generated in [16]. In [16], for the best results, some portion of the training time must be spent on learning the distribution of the entire image (without patches). Then during generation, the entire image is used as an input to the network. However, the goal for our paper was to avoid needing to input the entire image into the network both at training time and generation. Therefore, when running [16] in Figure 4, we trained the network only on patches of images and we generated full size images by first generating patches of the images (with positional encoding information) and then simply stitching them together.
>
> **Comment: Did the authors try using [23] in an unsupervised manner**
>
> When using [23] in an unsupervised manner, it is necessary to first train an unsupervised diffusion model (the patch-based networks from our work suffices for this) and then apply the unsupervised network to solve the inverse problem. The original paper is able to use the conditional network to enforce data consistency with the measurement, but with an unsupervised network, it is necessary to add in an additional step in the inference loop that enforces data consistency. We ran experiments using DPS as the data consistency strategy that is consistent with PaDIS. Further note that [23] has an additional tunable parameter which is the amount of overlap between patches.
>
> Table 6 shows the results of this method when this parameter has been tuned under the name Patch Averaging. The table shows the method can obtain reasonable results but is outperformed by our proposed method. Additionally, the optimal overlap parameter value requires a significant amount of overlap between patches (approximately 1/4 of the patch dimension in both x and y) which increases the number of patches per iteration whose score function must be evaluated using the network. Finally, while the empirical results are reasonable, there is a lack of mathematical justification in [23] for the procedure of averaging the predicted noises of overlapping patches and future work is required to theoretically justify (from a probability distribution perspective) this method. In the revision, we will include more visual examples of this method compared with the others.
>
> **Comment: Inference time comparisons should be demonstrated with other methods**
>
> Algorithm 1 indeed indicates that none of the sampling steps are skipped. For a fair comparison with other diffusion model based methods in Table 5, we used the same number of steps (1000) for all the methods; an increase in image quality was present with an increased number of steps for all the methods. DPS requires more time per image due to the computation of the gradient of the norm term which involves backpropagating through the network, and predictor-corrector sampling involves two network evaluations per iterations. The average reconstruction time per image in seconds for the methods in Tables 1 and 5 are shown below for 20 view CT.
>
> Baseline: 0.1
>
> ADMM-TV: 0.7
>
> Whole image diffusion: 172
>
> PaDIS (VE-DPS): 195
>
> Langevin dynamics: 98
>
> Predictor-corrector: 189
>
> VE-DDNM: 105
>
> **Comment: Visual results should be presented as a high resolution image**
>
> The global rebuttal PDF page has some examples of higher resolution images for different CT reconstruction experiments including 60 view reconstruction and fan beam CT. These images are displayed with higher contrast and should be more helpful in performing clinical diagnosis.

---

> > ### Comment · Reviewer_wfxx · 2024-08-13
> > **response to rebuttal**
> >
> > Thanks to the authors for their rebuttal and detailed comments. Majority of my concerns are answered, and I believe this submission went through a successful rebuttal period. Thus, I also increased my score.
> >
> > Please include the discussions presented here in the revised PDF as well, particularly the inference time comparisons that are provided here. It presents a more fair comparison of the proposed algorithm.

---

> > > ### Author Response · Authors · 2024-08-13
> > >
> > > Thank you for the review and reading our rebuttal. We will make sure to include these discussions in the revised paper. Feel free to let us know if there are any remaining questions about the manuscript and we will try our best to answer.

---

### Official Review · Reviewer_L95h · 2024-07-04

**Soundness:** 3
**Presentation:** 2
**Contribution:** 2
**Rating:** 5
**Confidence:** 5

**Summary:**

In this work, the authors propose a novel method for learning efficient data priors for entire images by training diffusion models only on image patches. During inference the authors introduce a patch-based position-aware diffusion inverse solver, which obtains the score function for the whole image through scores of individual patches and their positional encoding, using this as the prior for solving inverse problems. Multiple experiments on CT data as well as on the CELEBA dataset are conducted.

**Strengths:**

The idea of including positional information of patches into the diffusive reconstruction process seems novel and promising. Indeed, the authors demonstrate that their proposed approach can compute the score function for entire images without needing to feed the whole image through the network.

**Weaknesses:**

While the patch-based reconstruction for diffusion models appears promising, the evaluation and experiments presented in the paper require further extension to justify publication. Additional evaluations should be conducted against other models, reconstruction methods, inverse problems, and datasets. Furthermore, a sensitivity analysis should be included to examine how the proposed approach performs with different forward operators. Additionally, many of the presented CT reconstructions exhibit significant hallucinations, which is particularly concerning in the context of medical imaging. Overall, my decision is influenced by the weak evaluation of the method.

**Questions:**

I would be interested as to which number of data-samples the patch-based approach outperforms the Vanilla one?
How would the proposed approach perform with different forward operators, i.e. different inverse problems on the same datasets?
Why do the presented CT reconstructions exhibit such significant hallucinations, and how can this issue be addressed?

**Limitations:**

Although sparsely addressed in the conclusions, the authors do not give limitations about the proposed approach at all.
Especially, the proposed approach seems to have problems concerning hallucinations in the reconstructed CT images.

---

> ### Author Rebuttal · Authors · 2024-08-06
>
> We thank the reviewer for their insightful comments.
>
> **Comment: Additional evaluations should be conducted against other inverse problems, sensitivity analysis should be included with different forward operators**
>
> We conducted more experiments with different forward models: namely 60 view parallel beam CT, 180 view fan beam CT, and deblurring with a larger kernel of size 19x19. The results are shown in Table 7 and further demonstrate that our proposed method outperforms various SOTA methods for a large variety of forward models. Additionally, Table 6 contains a comparison with a wider variety of inverse problem solving methods. These comparisons with several other SOTA methods strengthen the evaluation of our method.
>
> **Comment: CT reconstructions exhibit significant hallucinations**
>
> The authors acknowledge that the images obtained by the generative models investigated including the proposed method for 20 view CT reconstruction show some hallucinations and artifacts. This is a natural consequence of using extreme compressed sensing with ultra-sparse views: normally, to reconstruct a 256x256 image requires (pi/2*256)=402 views, so for the 20 view experiments, the measurements have been compressed by a factor of 20. Due to this lack of information, it is very hard for any model to perform a diagnostic-quality reconstruction, though our proposed method (and the other diffusion model methods, to a lesser extent) are able to partially fill in this information through learning a strong image prior. The alternative methods that do not learn a prior perform significantly worse in terms of the shown metrics and exhibit severe blurring and artifacts. In clinical settings, it is much more common to perform patient diagnosis with CT scans consisting of hundreds of views. To illustrate this point, we perform experiments with 60 view CT, where our proposed method is able to obtain excellent quality images as shown in Figure B.1: essentially no artifacts are visible. (We show the potential of our proposed method to reconstruct images with ultra-sparse views with a decent image quality, which can be potentially used for other clinical applications such as patient positioning.)
>
> **Comment: What is the number of samples for which the proposed method is better**
>
> When looking at PSNR, the number of samples out of the 25 test samples in which the patch-based approach outperformed the vanilla one is as follows: 23 for 20 view CT, 25 for 8 view CT, 20 for deblurring, 16 for superresolution.  We will add this information to the supplement.
>
> **Comment: Authors do not give limitations of the proposed method**
>
> One limitation of the proposed approach (and all diffusion approaches) is that they tend to be slower than optimization based approaches, plug and play methods, and model-based learning methods. Another limitation of generative modeling approaches is the potential to hallucinate, particularly when the measurements are very compressed and contain little information. This is a limitation of most generative models, as illustrated by the visual examples from the whole image diffusion model for sparse view CT. This problem can be resolved by obtaining more projection views in a CT scan, depending on the application needs.

---

> > ### Comment · Reviewer_L95h · 2024-08-12
> > **Response**
> >
> > After considering the author's response, some of my concerns have been alleviated, prompting me to adjust my score to a borderline accept.

---

> > > ### Author Response · Authors · 2024-08-12
> > >
> > > Thank you for the review and reading our rebuttal. Your feedback is crucial for us to improve our manuscript. Feel free to let us know if there are any remaining questions about the manuscript and we will try our best to answer.

---

### Official Review · Reviewer_P2vb · 2024-07-10

**Soundness:** 2
**Presentation:** 2
**Contribution:** 3
**Rating:** 5
**Confidence:** 5

**Summary:**

This paper proposes a patch-based diffusion model for inverse problems, such as CT reconstruction and natural image deblurring. This method divides images into patches, reducing the size of the input data fed into the network, thereby decreasing memory consumption.

**Strengths:**

This method provides a feasible approach for dividing and merging patches to reduce boundary artifacts for diffusion models.

**Weaknesses:**

1. The innovation is limited. The main contribution of this paper is the application of patch diffusion [1] to inverse problems. However, the experimental results are neither promising nor convincing.

[1] Wang, Zhendong, et al. "Patch diffusion: Faster and more data-efficient training of diffusion models." Advances in neural information processing systems 36 (2024).

2. The CT images presented by the author in the paper and appendix are very blurry and not displayed with the correct window level and width, making it impossible to discern imaging details and lacking clinical significance.

3. Even though the CT images are very blurry, it is still evident that all the reconstructed CT images exhibit significant numbers of image artifacts (incorrect organ structures) compared to the ground truth. This is completely unacceptable for medical images.

4. The comparison methods are very limited, with only ADMM-TV, which is a very old method. There are numerous methods for CT reconstruction [2-4], natural image deblurring [5] and super-resolution [6] that the author did not compare with at all.

[2] Shen, Liyue, John Pauly, and Lei Xing. "NeRP: implicit neural representation learning with prior embedding for sparsely sampled image reconstruction." IEEE Transactions on Neural Networks and Learning Systems 35.1 (2022): 770-782.

[3] Wu, Qing, et al. "Self-supervised coordinate projection network for sparse-view computed tomography." IEEE Transactions on Computational Imaging 9 (2023): 517-529.

[4] Chung, Hyungjin, et al. "Solving 3d inverse problems using pre-trained 2d diffusion models." Proceedings of the IEEE/CVF Conference on Computer Vision and Pattern Recognition. 2023.

[5] Tang, Xiaole, et al. "Uncertainty-aware unsupervised image deblurring with deep residual prior." Proceedings of the IEEE/CVF conference on computer vision and pattern recognition. 2023.

[6] Wang, Longguang, et al. "Unsupervised degradation representation learning for blind super-resolution." Proceedings of the IEEE/CVF conference on computer vision and pattern recognition. 2021.

**Questions:**

1. I suggest the authors make more improvements for image restoration tasks (inverse problems) to control the reliability of the generated images and reduce artifacts.
2. it is necessary to compare with more SOTA methods to obtain a more comprehensive evaluation.

**Limitations:**

1. Due to the severe artifacts observed in the visual results (CT reconstruction) and the limited number of comparison methods, the experimental results are hardly convincing.
2. The innovation is very limited. It is based on existing diffusion models and merely applied to the domain of inverse problems (image restoration tasks).

---

> ### Author Rebuttal · Authors · 2024-08-06
>
> We thank the reviewer for their insightful comments.
>
> **Comment: Innovation is limited**
>
> The main innovation of our method is a method to formulate a diffusion based image prior from solely the patches of the image. Diffusion models are known for requiring a large amount of memory for training and inference and extending them for large scale images is a challenging problem. This work illustrates how image priors for very large images can be learned by learning priors of patches, a much less memory intensive task. This is unlike the work of [1], which ultimately still requires training on whole images as well as inputting the whole image into the network to generate images. We also demonstrate in Table 3 that our proposed method can learn a reasonable prior for dataset sizes much smaller than is normally used for training diffusion models and the advantage over whole image diffusion models becomes more pronounced for small datasets. Finally, unlike previous patch-based diffusion model papers such as [16] and [17] that can only be used for generating images, we show how our method learns a full image prior from only image patch training that can be coupled with most diffusion inverse problem solving algorithms, which is not a trivial extension based on [16] and [17].
>
> **Comment: CT images are blurry and displayed with the wrong window level**
>
> We displayed CT images corresponding to our new CT experiments in Figure B.1 using a narrower window size of 800 to 1200 HU and with higher resolution. These images show better contrast between different organs and are more useful for obtaining a clinical diagnosis. In the revision, we will redisplay all the CT reconstruction images with this higher level of contrast.
>
> **Comment: CT images contain significant number of image artifacts**
>
> The authors acknowledge that the images obtained by the generative models investigated including the proposed method for 20 view CT reconstruction show some hallucinations and artifacts. This is a natural consequence of using extreme compressed sensing with ultra-sparse views: normally, to reconstruct a 256x256 image requires (pi/2*256)=402 views, so for the 20 view experiments, the measurements have been compressed by a factor of 20. Due to this lack of information, it is very hard for any model to perform a diagnostic-quality reconstruction, though our proposed method (and the other diffusion model methods, to a lesser extent) are able to partially fill in this information through learning a strong image prior. The alternative methods that do not learn a prior perform significantly worse in terms of the shown metrics and exhibit severe blurring and artifacts. In clinical settings, it is much more common to perform patient diagnosis with CT scans consisting of hundreds of views. To illustrate this point, we perform experiments with 60 view CT, where our proposed method is able to obtain excellent quality images as shown in Figure B.1: essentially no artifacts are visible. (We show the potential of our proposed method to reconstruct images with ultra-sparse views with a decent image quality, which can be potentially used for other clinical applications such as patient positioning.)
>
> **Comment: Comparison methods are very limited**
>
> [4] is a method that applies 2D diffusion models to solve 3D inverse problems including CT reconstruction, whereas our proposed method applies for 2D inverse problems, so it cannot directly be applied. However, the sampling algorithm that is used is the predictor-corrector sampling algorithm, which we initially compared to in Table 5 and now added a more comprehensive comparison in Table 6. This sampler did not perform as well as the one chosen for PaDIS (DPS) by the quantitative metrics.
>
> [2] and [3] are self-supervised methods for CT reconstruction, which means that network is trained during reconstruction time, substantially slowing down the algorithm. [5] is a deep image prior method that also requires network training at inference time. Furthermore, [2], [3], [5], and [6] are all problem specific methods for which a generalization to the other types of inverse problems would be nontrivial. Our proposed method, along with most of the methods we compared to in Table 6, are easily generalizable methods that can solve a wide variety of inverse problems. Furthermore, the training process for our algorithm need only happen once per dataset, and no network training is required during reconstruction time. Due to these fundamental differences between our method and [2], [3], [5], [6], we believe that comparisons with those methods would not be fair.
>
> Nevertheless, to provide a more complete evaluation of our method, we included additional comparisons with plug and play (PnP) methods and other diffusion inverse solvers in Table 6. These methods share the similarity with our method in that network training only needs to be done once per dataset, and the same trained network can be used for different types of inverse problems, allowing for greater flexibility. Table 6 consists of an expanded comparison between various methods. We implemented various diffusion inverse solving methods [1], [7], [19] in conjunction with our patch-based prior. We included two additional patch based methods from [23] and [69] where we applied [23] in an unsupervised way by using the same unsupervised network trained in our proposed method and adding a DPS step during reconstruction. We also implemented two plug and play (PnP) methods by first training denoisers on CT images and the CelebA dataset and then applying these denoisers in an unsupervised way to solve the inverse problems. Optimal hyperparameters for all these methods were found through searching.
>
> In all cases, our proposed method outperformed the comparison methods. In the revision, we will add visual examples of these methods. These comparisons with several other SOTA methods strengthen the evaluation of our method.

---

> > ### Comment · Reviewer_P2vb · 2024-08-12
> > **The rebuttal response**
> >
> > The author's rebuttal addressed some of my concerns, so I have revised my score to borderline accept. However, there are still some unresolved issues, as outlined below.
> >
> > 1. The authors directly addressed the issue of CT artifacts, and I acknowledge their explanation that diffusion-based extremely-limited-angle CT reconstruction inherently results in such artifacts. However, from a medical application perspective, these artifacts are quite concerning. In the rebuttal, the author adds reconstruction results from 60 angles. However, since the selected images primarily show the lungs (which appear as zero in the specified window width), they do not contain enough tissue, unlike abdominal images, to adequately assess the reconstruction quality.
> >
> > 2. In both the paper and the rebuttal, the author  points out that performing diffusion on patches can reduce computational costs, enabling to process large-scale images, such as higher resolution and 3D images. In the experiments, both the CT dataset and the CelebA-HQ dataset are 256x256. CelebA-HQ is a standard dataset, but CT data often comes in much higher resolutions, such as 512x512. I suggest the authors conduct experiments on such CT datasets to better support their claims.
> >
> > 3. In summary, while the author's method performs well in terms of quantitative metrics, the inevitable artifacts in the CT data raise some concerns from a medical perspective.

---

> > > ### Author Response · Authors · 2024-08-13
> > >
> > > **Comment: Selected images show the lungs and do not contain enough tissue unlike abdominal images**
> > >
> > > The requirements of the rebuttals state that we cannot use links in any part of the response except for code (and we can no longer modify our one page PDF), but we have sent a message to the AC asking if it would be permissible to share an anonymized link to images of our new results. We have run experiments on CT images containing more tissue and contrast demonstrating that for 60 view CT, our method is able to obtain high quality reconstructions which do not exhibit artifacts.
> > >
> > > **Comment: CT data comes in 512x512 resolution, I suggest authors conduct experiments on such CT datasets**
> > >
> > > The original AAPM dataset cited in the paper consists of 512x512 images. We used this original data scaled between 0 and 1 in the same way as the 256x256 CT images in previous experiments to train a patch based network. The largest patch size was chosen to be 64x64, while patches of size 32x32 and 16x16 were also used for training. The zero padding was set to 64 pixels on all four sides of the image. Due to time constraints, we were only able to train the network for roughly 20 hours.
> > >
> > > For reconstruction, only the largest patch size was used: a total of 81 patches of size 64x64 were needed to fully cover the 512x512 image while allowing for shifts. Similarly we cannot show visual results of the reconstruction, so we report the quantitative results of the 60 view parallel beam CT reconstruction problem: over the test dataset, the average PSNR was 36.92 and the average SSIM was 0.899. This shows that the patch prior was learned well and leads to a high quality reconstruction free of artifacts. In the revision, we will provide a more complete comparison of applying various methods on these higher resolution CT images as well as more visual results.
> > >
> > > **Comment: Inevitable artifacts in CT data raise some concerns from medical perspective**
> > >
> > > We acknowledge that artifacts may arise for very sparse view CT reconstructions. In clinical settings, hundreds of views are typically used to perform patient diagnoses. Our experiments on 60 view CT show a lack of artifacts, so in the future, our proposed method could be used to reduce the number of views needed to obtain an accurate reconstruction for medical settings.

---

### Official Review · Reviewer_YkyD · 2024-07-11

**Soundness:** 3
**Presentation:** 3
**Contribution:** 3
**Rating:** 7
**Confidence:** 3

**Summary:**

This manuscript discusses diffusion models for inverse problems. The authors discuss using image patches of the image to improve computational bottlenecks and overcoming the lack of sufficient data in training appropriate surrogate neural network priors for the inversion task. The authors discuss details of their proposed method and illustrate the advantages of their methods on various tasks including CT, deblurring, and superresolution.

**Strengths:**

This manuscript is well-written and structured, making it easy for readers to follow the presented ideas. The authors ground their work in existing literature and reference relevant papers in this field.  Data-driven approaches for inverse problems have shown significant advances and this manuscript contributes to this field.

**Weaknesses:**

This work is partly incremental and heavily relies on various previous and cited publication, e.g., [12,18,19]. Furthermore,  I assume computational costs for the solution of the inverse problem are extremely large since just a stochastic gradient approach must be utilized in the inversion process (see algorithm 1). These are typically prohibitive for large-scale inverse problems removing the advantages of the learned data-driven prior.

**Questions:**

See Weaknesses.

**Limitations:**

Yes

---

> ### Author Rebuttal · Authors · 2024-08-06
>
> We thank the reviewer for their insightful comments.
>
> **Comment: Work is partly incremental, heavily relies on cited publications**
>
> The papers [12] and [18] apply diffusion models to solve 3D reconstruction problems, whereas the proposed method performs experiments on 2D reconstruction problems using a patch-based prior. [19] uses the predictor-corrector method for solving medical imaging problems, which we compared to in Table 5. We revised the labels in Table 5 to make this more clear.  The method of diffusion inverse problem solving most closely related to our method is DPS [5]. However, the most significant contribution of the proposed method is a patch-based image prior that requires only patch inputs to a neural network and can be paired with any diffusion inverse solving algorithm, as illustrated in Table 5.
>
> **Comment: Computational costs are large and requires a stochastic gradient approach**
>
> We acknowledge that the computational costs of the proposed method will exceed that of more traditional reconstruction methods, as is true for almost all diffusion based methods, as a tradeoff for achieving better image quality. However, the computational cost of the proposed method is similar to other diffusion based methods. The average reconstruction time per image in seconds for the methods in Tables 1 and 5 are shown below for 20 view CT. Notably, our proposed method takes only slightly more time than the approach using diffusion models trained on entire images while greatly reducing the memory needed and improving the result.  We will add this table to the supplement.
>
> Baseline: 0.1
>
> ADMM-TV: 0.7
>
> Whole image diffusion: 172
>
> PaDIS (VE-DPS): 195
>
> Langevin dynamics: 98
>
> Predictor-corrector: 189
>
> VE-DDNM: 105
>
> The stochastic approach taken in Algorithm 1 allows the runtime of the algorithm to be similar to other diffusion methods while eliminating artifacts that would otherwise persist between boundaries of patches. The approach is similar to the one taken in the paper below: instead of computing the score function multiple times each iteration, stochastically choose one of them to compute each iteration. Over the course of hundreds of iterations throughout the reconstruction process, the stochastic approximation of the score function becomes more accurate. Furthermore, this approach does not sacrifice the advantages of using this data-driven prior, as Figure 4 shows that our proposed method can still be used to unconditionally generate fairly realistic images.
>
> S. Lee, H. Chung, M. Park, J. Park, W.-S. Ryu, and J. C. Ye. “Improving 3D imaging with pre-trained perpendicular 2D diffusion models”. In: Proceedings of the IEEE/CVF International Conference on Computer Vision. 2023, pp. 10710–10720.

---

> > ### Comment · Reviewer_YkyD · 2024-08-12
> >
> > Thanks a lot for your response. While some of the author's comments address my concerns, I will maintain my initial ratings.

---

> > > ### Author Response · Authors · 2024-08-12
> > >
> > > Thank you for the review and reading our rebuttal. Your feedback is crucial for us to improve our manuscript. Feel free to let us know if there are any remaining questions about the manuscript and we will try our best to answer.

---

### Official Review · Reviewer_mcvX · 2024-07-13

**Soundness:** 3
**Presentation:** 3
**Contribution:** 2
**Rating:** 6
**Confidence:** 3

**Summary:**

The authors present an approach for tile-based training and prediction of diffusion models applied for inverse problem posterior sampling.
The core idea is that training is done with random patches and during generation the authors use a differently shifted non-overlapping tiling grid for each iteration of the process.
The authors additionally provide the x- and y-pixel-coordinates to the network by encoding them in extra channels and concatenating them.
Their approach allows them to generate images without Stichting artefacts.
The random tiling during training can be seen as data augmentation and the authors show that this enables their method to be trained on smaller datasets.

**Strengths:**

* Improving the memory requirement for diffusion model is an important problem. Many applications need to process large images in a coherent way, while avoiding stitching artefacts.
* I appreciate the fact that the proposed training scheme enables training with less data. And that the authors validate this in an experiment.
* The authors show that their method does not depend on the particular smapling scheme or network architecture. I appreciate the generality of the approach.

**Weaknesses:**

* My main criticism is regarding the motivation of the problem.
The authors write:
"*Directly using overlapping patches would result in sections of the image covered by multiple patches to be updated multiple times, which is inconsistent with the theory of diffusion models.*" The question of how tiling and stitching can be applied for unets, such that the result is equivalent to processing the image as a whole has been explored before. See questions section for details.

* The proposed tiling comes at a cost: It limits the range of correlations that can be captured by the diffusion model.
This is visible in Figure 4, where the generated images show no stitching artefacts, but also produce nonsensical large scale anatomy.
I am missing a discussion of this aspect.
In the posterior samples, the effect is not visible, since the input image contains enough information such that long range correlations are not relevant.
I believe this would be a problem for inverse problems where the input image contains less information, i.e., when the noise is very severe, or for super resolution with a more extreme resolution factor.

**Questions:**

It is not correct that using overlapping tiles would be "*inconsistent with the theory of diffusion models*".
In general, a tiling scheme for unets can be implemented, such that the stitched tiles are identical to the result of processing the image as a whole.
This can be achieved by using overlapping tiles (with the correct shift) and disregarding the areas close to the border in the outputs that are influence by padding. I believe a discussion of this can be found in [1].
What would prevent us from applying this approach with a diffusion model in each step? The network could still be trained using patches.

While such a tiling approach would cost additional computation time, since it requires overlapping patches, it should theoretically produce guaranteed stitch artefact-free outputs.
Do the authors agree, that this well established approach should ideally be a baseline or at least be discussed?
The authors could show that their method produces comparable results at reduced computational cost.

[1]: Rumberger, Josef Lorenz, et al. "How shift equivariance impacts metric learning for instance segmentation." Proceedings of the IEEE/CVF International Conference on Computer Vision. 2021.

**Limitations:**

I am missing a discussion on the cost of the proposed patch scheme regarding the ability of the network to model long range correlations. See weaknesses.

---

> ### Author Rebuttal · Authors · 2024-08-06
>
> We thank the reviewer for their insightful comments.
>
> **Comment: Tiling scheme for unets can be implemented with overlapping patches**
>
> We implemented the method provided in the above reference [1] while using the same trained network, hyperparameters, and DPS for inverse problem solving. The network consisted of 3 layers of upsampling/downsampling and each layer involved pooling with a factor of 2. The largest patches that the network was trained on had size 56. To satisfy the hypotheses of the paper, the overlap between patches was to be a multiple of 8; we set it equal to 8 to minimize the number of patches for image partitioning. Hence each 256x256 image was divided into 36 overlapping patches, compared to 25 for PaDIS.
>
> Table 6 shows the results of using this method under the name Patching Stitching. For the shown inverse problems, the method obtains reasonable results but performs worse than our proposed method. Despite this, the reconstructed images did not appear to exhibit any boundary artifacts along patch boundaries, showing that the method of [1] worked in that aspect. The increased number of patches necessary for that approach increased the runtime of the reconstruction algorithm by approximately 30%.
>
> We also examined the unconditionally generated images using the method of [1] which are shown in Figure B.2. Although these images exhibited relatively smooth features without any clear boundary artifacts between patches (unlike the clear artifacts visible in the middle row of Figure 4 generated by naive patch stitching), the overall structure was highly inconsistent with the CT images in the dataset. This is in contrast with the images generated by PaDIS as shown in the bottom row of Figure 4. Therefore, although [1] can result in smooth images, the lack of patch shifting means that the learned image prior differs from the one in Eq. (3), resulting in unrealistic looking generated images, which indicated the underlying data distribution cannot be well captured. The significantly worse prior learned by this method is reflected in the worse results for inverse problem solving and especially when the measurements are highly compressed.
>
> There are two aspects of the UNet used in our application that likely caused that patch stitching method to fail to learn the prior well. Firstly, our diffusion UNet takes in an additional scalar input indicating the noise level of the noisy image being input into the network. This scalar input is processed through a sinusoidal positional encoding before being embedded into the layers of the UNet through an attention mechanism. Secondly, our network takes in the positional embedding of the location of the patch via concatenation along the channel dimension. Thus, our network learns the score function of patches while also incorporating the location of the patch, making it different from traditional UNets used for segmentation [1].
>
> **Comment: Patches limit the ability of the network to learn long range correlations**
>
> Learning longer range correlations within an image is assisted by the method of using positional encoding of patches as an input to the network: it allows our network to learn a different distribution of patches at different locations in the image. Thus, provided that the location of the central object to be imaged is in a relatively consistent location (as is the case for CT scans or human faces), the network can learn that, for instance, the spine of the CT scan is typically around the middle bottom section of the image. Such learning is consistent with the generated results in Figure 4. Nevertheless, generating whole images with realistic large scale anatomy is challenging, and we show the generation results to demonstrate that they appear somewhat reasonable while emphasizing the focus on solving inverse problems.
>
> Figure 5 contains some examples of reconstructed images from 8 view CT, a very compressed sensing problem. Normally, to reconstruct a 256x256 image would require (pi/2*256)=402 views, so the compression is a factor of 50. In this case, the images show reasonable large scale anatomy.

---

> > ### Comment · Reviewer_mcvX · 2024-08-12
> > **Thanks for the rebuttal - additional questions.**
> >
> > Thank you for the rebuttal.
> >
> > **Regarding the tiling and Stichting method:**
> > I appreciate that you try the suggested stitching mechanism and put results into the pdf.
> > The sampled results look indeed inferior.
> > You write:
> >
> > *"There are two aspects of the UNet used in our application that likely caused that patch stitching method to fail to learn the prior well. Firstly, our diffusion UNet takes in an additional scalar input indicating the noise level of the noisy image being input into the network. This scalar input is processed through a sinusoidal positional encoding before being embedded into the layers of the UNet through an attention mechanism. Secondly, our network takes in the positional embedding of the location of the patch via concatenation along the channel dimension. Thus, our network learns the score function of patches while also incorporating the location of the patch, making it different from traditional UNets used for segmentation [1]. "*
> >
> > I agree that these are the likely reasons for the performance difference. I did not suggest the tiling strategy as an alternative to providing noise level and patch positions as additional inputs (I think these make a lot of sense), but rather as the established method to avoid stitching artefacts, as opposed to the proposed method of using different shifts in each iteration of the generation process. I don't see a reason not to combine the established way of avoiding stitching artefacts in UNets with the additional input information (position and noise level)
> >
> >
> >
> > **Regarding long range correlations:**
> > I believe learning *"different distribution of patches at different locations in the image"* is not the same as learning long range correlations, which means learning which structures on the one side of the image are likely to appear together with structures on the other side of the image. I still don't think it is possible to learn such correlations without looking at the image as a whole.

---

> > > ### Author Response · Authors · 2024-08-12
> > >
> > > Thanks for the review and reading our rebuttal. Your feedback is crucial for us to improve our manuscript.
> > >
> > > **Comment: I did not suggest the tiling strategy as an alternative to providing noise level and patch positions as additional inputs**
> > >
> > > In our experiments of using tiling UNets in Table 6 and Figure B.2, we still included the noise level and patch positions as additional inputs: for a fair comparison, we used the **same trained network** that was used to reconstruct the CT images of Figure 5. The only difference was that at reconstruction time, instead of using shifting non-overlapping patches, we used fixed location overlapping patches with the tiling UNet strategy. The patch size at reconstruction time (56x56) was kept the same for the old experiments (Figure 5) and new experiments (Patch Stitching in Table 6 and Figure B.2).
> > >
> > > Note that in Figure B.2, although the generated images are of significantly worse quality than the bottom row of Figure 4 (the proposed method), the overall shape of the CT images are still preserved and the spine is roughly located in the correct position for all the images. This is due to the position encoding inputs to the network: if these inputs were not included, the network would learn a **mixture** of distributions consisting of patches of all different locations, and it would be impossible for the network to “know” that the spine should be at the central bottom area of the image. Hence, the positional input is crucial for obtaining even somewhat reasonable looking generated images.
> > >
> > > In the rebuttal, we highlight the difference between the network used in the generation of Figure B.2 and the types of networks studied in [1]. Although the network used for Figure B.2 utilizes the same ideas as in [1], the two additional inputs (namely, the noise level input and positional encoding) may create additional complications especially in terms of the formulation of the prior as a product of patch priors. That the generated images of Figure B.2 show no discontinuities between boundaries of patches is an indication that while the goal of eliminating boundary artifacts is achieved by the method of [1], the underlying learned prior is worse than our proposed method.
> > >
> > > **Comment: I still don’t think it is possible to learn correlations without looking at the image as a whole**
> > >
> > > We acknowledge that the proposed method would not be able to learn which structures on one side of the image are likely to appear with structures on the other side of the image. It would only be able to learn that independently, certain types of structures may be likely to appear on one side, whereas other types of structures may be likely to appear on the other side, but not be able to learn a connection between them. This is a limitation of using a network that only accepts patches as inputs, but we demonstrate that this learned prior is sufficient for solving inverse problems (the focus of this work) particularly when data is limited.

---

### Author Rebuttal · Authors · 2024-08-06

We would like to sincerely thank all the reviewers for the valuable comments and constructive feedback on our paper. We provide point-by-point responses to address each reviewer’s comments and highlight our response to some key questions and additional experiments and results as below:

**More baselines for comparison**: We provided new results to compare with more baseline methods in Table 6 as shown in the attached pdf, including three diffusion-based methods [1,7,19], two plug and play (PnP) methods [42, 46], and two patch-based methods [23, 69] as suggested by Reviewers 2 and 3. This comprehensive comparison shows that our proposed method can outperform these relevant methods to a large extent by learning a better image prior and applying the optimal inverse solving algorithm.

**Different forward operators**: We conduct more experiments with different forward models: namely 60 view parallel beam CT, 180 view fan beam CT, and deblurring with a larger kernel of size 19x19. The results are shown in Table 7 and further demonstrate that our proposed method outperforms various SOTA methods for a large variety of forward models.

**Window size**: We displayed CT images corresponding to our new CT experiments in Figure B.1 using a narrower window size of 800 to 1200 HU. These images show better contrast between different organs and are more useful for obtaining a clinical diagnosis.

**Hallucinations and artifacts in the reconstructed images**: The presence of artifacts in some of the reconstructed CT images using generative methods is a natural consequence of using extreme compressed sensing with ultra-sparse views. Due to this lack of information, it is very hard for any model to perform a diagnostic-quality reconstruction, though our proposed method performs best in terms of the quantitative metrics. We added experiments with 60 view CT, where our proposed method is able to obtain excellent quality images as shown in Figure B.1: essentially no artifacts are visible.

**Innovation**: The main innovation of our method is a method to formulate a diffusion based image prior from solely the patches of the image. This work illustrates how image priors for very large images can be learned by learning priors of patches, a much less memory-intensive and data-hungry task. Unlike previous patch-based diffusion model papers such as [16] and [17] that can only be used for generating images, we show how our method learns a full image prior from only image patch training that can be coupled with most diffusion inverse problem solving algorithms, which is not a trivial extension based on [16] and [17].

[69]: Rumberger, Josef Lorenz, et al. "How shift equivariance impacts metric learning for instance segmentation." Proceedings of the IEEE/CVF International Conference on Computer Vision. 2021.

---

### Decision · Program_Chairs · 2024-09-25

**Decision:**

Accept (poster)

**Comment:**

Proposes a novel approach to scaling diffusion-based generative models to high-resolution images, by modeling randomly-tiled patches of images, which is motivated in part by reducing memory and training-data demands.  An acknowledged limitation is that the resulting generative model cannot capture very-distant correlations, but it is still effective for solving inverse-problems as focused on in experiments on both natural and medical images.  Reviews voiced several concerns about experimental details and baselines, which were mostly addressed by a thorough author rebuttal.  Please ensure that these additional results and clarifications are included in your final manuscript.